# A cell atlas of the chick retina based on single-cell transcriptomics

**Masahito Yamagata†, Wenjun Yan†, Joshua R Sanes***

Center for Brain Science and Department of Molecular and Cellular Biology, Harvard University, Cambridge, United States

**Abstract** Retinal structure and function have been studied in many vertebrate orders, but molecular characterization has been largely confined to mammals. We used single-cell RNA sequencing (scRNA-seq) to generate a cell atlas of the chick retina. We identified 136 cell types plus 14 positional or developmental intermediates distributed among the six classes conserved across vertebrates – photoreceptor, horizontal, bipolar, amacrine, retinal ganglion, and glial cells. To assess morphology of molecularly defined types, we adapted a method for CRISPR-based integration of reporters into selectively expressed genes. For Müller glia, we found that transcriptionally distinct cells were regionally localized along the anterior-posterior, dorsal-ventral, and central-peripheral retinal axes. We also identified immature photoreceptor, horizontal cell, and oligodendrocyte types that persist into late embryonic stages. Finally, we analyzed relationships among chick, mouse, and primate retinal cell classes and types. Our results provide a foundation for anatomical, physiological, evolutionary, and developmental studies of the avian visual system.

***For correspondence:**
sanesj@mcb.harvard.edu

†These authors contributed equally to this work

**Competing interests:** The authors declare that no competing interests exist.

## Introduction

The retina is about as complex as other regions of the vertebrate central nervous system. It differs from many regions, however, in being particularly accessible to study. For example, its neurons can be imaged live without surgical intervention, visual stimuli can be precisely controlled in time and space, and the paucity of long-distance inputs facilitates analysis of the entire circuit ex vivo. These and other technical advantages, along with the intrinsic importance of the retina and the fact that most blinding diseases arise from retinal dysfunction, have combined to make the retina a popular model for analysis of neural structure, function, development, and disease (*Wässle, 2004*; *Dowling, 2012*; *Hoon et al., 2014*). Accordingly, retinas of many vertebrate species have been studied in detail, including those of rodents (e.g. mice and rats), carnivores (e.g. cats and ferrets), primates (e.g. macaques, marmosets, and humans), birds (e.g. chickens and pigeons), fish (e.g. zebrafish and goldfish), reptiles (e.g. turtles and lizards), and amphibia (e.g. salamanders and frogs) (*Dowling, 2012*; *Thoreson and Dacey, 2019*; *Baden et al., 2020*). All these studies depend on classification and characterization of the cell types that comprise the retina. Recently, this enterprise has been greatly enhanced by the introduction of methods for high-throughput single-cell transcriptomic profiling (scRNA-seq), which enable comprehensive and minimally biased sampling of cell types. To date, however, with the exception of a single study on zebrafish RGCs (*Kölsch et al., 2020*), these methods have been applied only to mice and primates (*Macosko et al., 2015*; *Shekhar et al., 2016*; *Rheaume et al., 2018*; *Liang et al., 2019*; *Menon et al., 2019*; *Peng et al., 2019*; *Tran et al., 2019*; *Cowan et al., 2020*; *Lu et al., 2020*; *Orozco et al., 2020*; *Yan et al., 2020a*; *Yan et al., 2020b*; reviewed in *Shekhar and Sanes, 2021*), restricting use of non-mammalian models and making it difficult to draw evolutionary relationships among species at the molecular level. Here, we address this limitation by using scRNA-seq to generate a cell atlas of the chick retina.

The basic plan of the retina is highly conserved among all vertebrates (*Baden, 2020*). Five neuronal classes are arranged in three cellular (nuclear) layers separated by two synaptic (plexiform) layers:

**eLife digest** The evolutionary relationships of organisms and of genes have long been studied in various ways, including genome sequencing. More recently, the evolutionary relationships among the different types of cells that perform distinct roles in an organism, have become a subject of inquiry. High throughput single-cell RNA sequencing is a technique that allows scientists to determine what genes are switched on in single cells. This technique makes it possible to catalogue the cell types that make up a tissue and generate an atlas of the tissue based on what genes are switched on in each cell. The atlases can then be compared among species.

The retina is a light-sensitive tissue that animals with a backbone, called vertebrates, use to see. The basic plan of the retina is very similar in vertebrates: five classes of neurons – the cells that make up the nervous system – are arranged into three layers. The chicken is a highly visual animal and it has frequently been used to study the development of the retina, from understanding how unspecialized embryonic cells become neurons to examining how circuits of neurons form. The structure and role of the retina have been studied in many vertebrates, but detailed descriptions of this tissue at the molecular level have been largely limited to mammals.

To bridge this gap, Yamagata, Yan and Sanes generated the first cell atlas of the chicken retina. Additionally, they developed a gene editing-based technique based on CRISPR technology called eCHIKIN to label different cell types based on genes each type switched on selectively, providing a means of matching their shape and location to their molecular identity. Using these methods, it was possible to subdivide each of the five classes of neurons in the retina into multiple distinct types for a total of 136.

The atlas provided a foundation for evolutionary analysis of how retinas evolve to serve the very different visual needs of different species. The chicken cell types could be compared to types previously identified in similar studies of mouse and primate retinas. Comparing the relationships among retinal cells in chickens, mice and primates revealed strong similarities in the overall cell classes represented. However, the results also showed big differences among species in the specific types within each class, and the genes that were switched on within each cell type.

These findings may provide a foundation to study the anatomy, physiology, evolution, and development of the avian visual system. Until now, neural development of the chicken retina was being studied without comprehensive knowledge of its cell types or the developmentally important genes they express. The system developed by Yamagata, Yan and Sanes may be used in the future to learn more about vision and to investigate how neural cell types evolve to match the repertoire of each species to its environment.

photoreceptors (PRs) in an outer nuclear layer (ONL), three sets of interneurons (horizontal, bipolar, and amacrine cells; HCs, BCs, ACs) in an inner nuclear layer (INL), and output neurons (retinal ganglion cells, RGCs), along with some ACs, in a ganglion cell layer (GCL; *Cajal, 1892*; *Masland, 2012*; *Figure 1A–B*). PRs form synapses with HCs and BCs in an outer plexiform layer (OPL), while RGCs, BCs, and ACs form synapses in an inner plexiform layer (IPL). Axons of RGCs then exit the eye and travel through the optic nerve to a variety of retinorecipient areas in the brain (*Dhande et al., 2015*; *Martersteck et al., 2017*). Each of these classes is divided into types with specific patterns of connectivity among them endowing distinct RGC types with sensitivities to different visual stimuli, such as edges, moving or oriented objects, and color contrasts (*Sanes and Masland, 2015*). The retina also contains glial cells: Müller glia in the INL, and in many species, astrocytes and oligodendrocytes in and beneath the GCL (*Reichenbach and Bringmann, 2013*; *Vecino et al., 2016*). Altogether, transcriptomic and morphological studies have identified a total of >130 neural (neuronal and glial) cell types in mice and ~70 in primates (*Yan et al., 2020a*; *Yan et al., 2020b*; *Peng et al., 2019*).

Although the basic retinal plan is conserved, cell types and patterns of connectivity vary among species, serving their visual needs (*Baden et al., 2020*). Birds are highly visual animals with sizable eyes, generally high acuity and sophisticated retinas (*Cook, 2000*; *Seifert et al., 2020*). For example, although most mammals have two cone PR types, most birds are tetrachromatic, with cone PRs selectively sensitive to red, green, blue, and ultraviolet light (*Hart, 2001*; *Baden and Osorio, 2019*). Many have regions specialized for high acuity vision, akin to the fovea found in primates. Indeed,

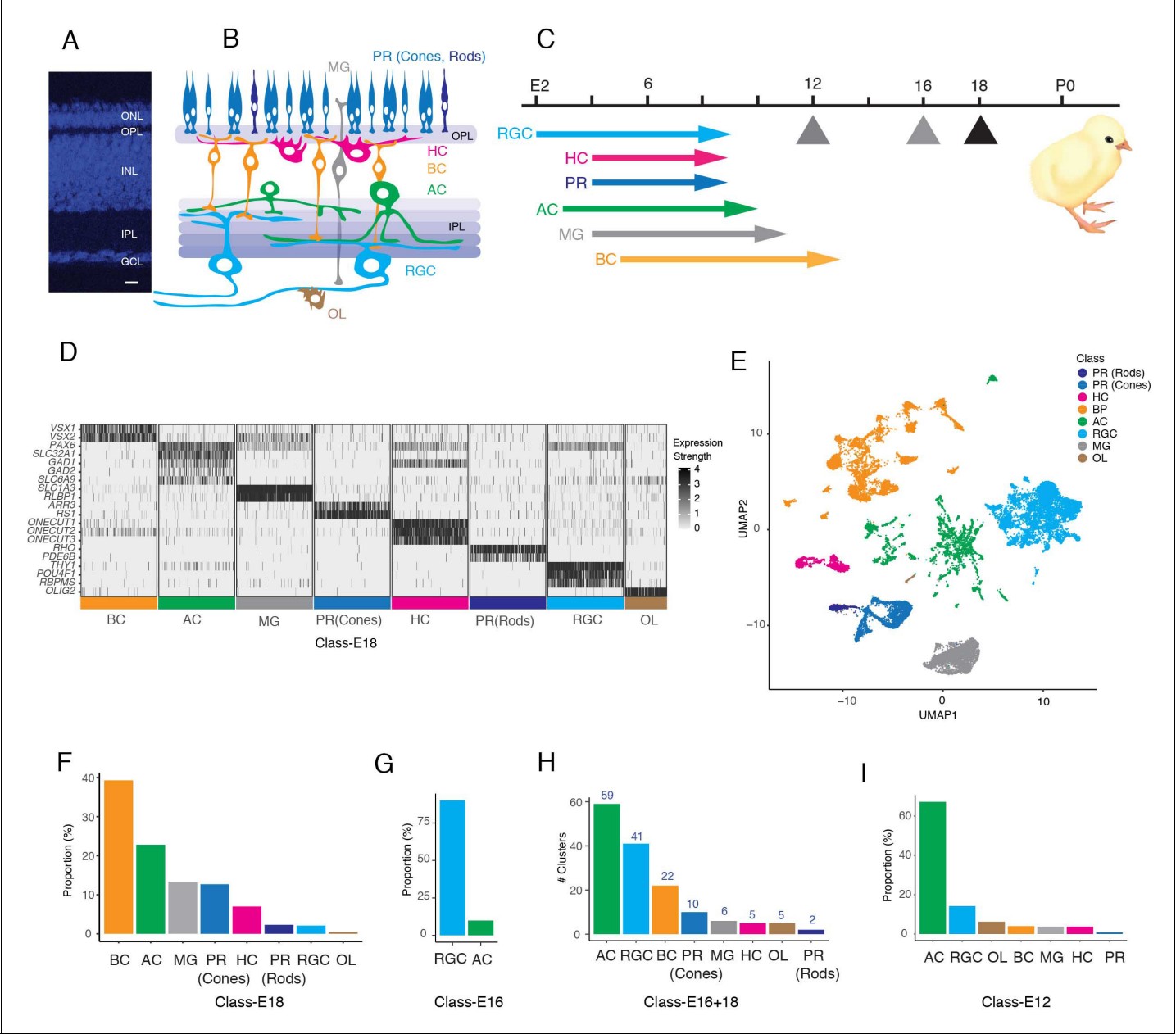

**Figure 1.** Datasets used to generate a chick retinal atlas. (**A**) Cross-section of chick retina stained with NeuroTrace to mark somata. The retina consists of three cellular layers: outer nuclear layer (ONL), inner nuclear layer (INL), and ganglion cell layer (GCL) separated by two synaptic layers, outer plexiform (OPL) and inner plexiform (IPL). Bar, 10 μm. (**B**) Sketch showing retinal cell classes. The ONL contains photoreceptors (PR): double cones, single cones, and rods. The INL contains horizontal, bipolar, and amacrine cells (HC, BC, and AC) and Müller glia (MG). The GCL contains retinal ganglion cells (RGCs) and ACs. Oligodendrocytes (OL) are present in an axonal layer beneath the GCL. (**C**) Birthdates of each class, from *Prada et al., 1991*. Hatching (P0) is at embryonic day (E)21. Arrows denote ages at which cells were obtained for scRNA-seq. To generate the cell atlas, E16 data were used for RGCs and E18 data for all other classes. (**D**) Expression of a subset of marker genes used to allocate E18 retinal cells to classes. Plot shows scaled expression level in a randomly down-sampled subset of all cells. (**E**) UMAP of E16+18 scRNA-seq data with class names based on D. (**F**) Fraction of E18 cells in each cell class, as determined by expression of canonical markers in D. (**G**) Fraction of E16 RGC-enriched cells in each cell class, determined as in D, F. (**H**) Number of clusters (putative cell types) in each retinal cell class, based on reclustering each class separately. (**I**) Fraction of E12 RGC-enriched cells in each cell class, determined as in D, F.

The online version of this article includes the following figure supplement(s) for figure 1:

**Figure supplement 1.** Quality control metrics for datasets used in this paper .

histological and immunohistochemical studies have suggested that there may be more cell types in avian retinas than in those of mammals (*Cajal, 1892*; *Mariani and Leure-DuPree, 1977*; *Hayes, 1982*; *Quesada et al., 1988*; *Naito and Chen, 2004*; *Karten and Brecha, 1983*; *Brecha et al., 1984*; *Karten et al., 1990*).

Among birds, the retina of the domestic chicken (*Gallus gallus domesticus*) has been the most intensively studied. In particular, it has been a favored model for developmental analyses, including studies on the generation and migration of retinal neurons, their diversification into classes and types, the growth and guidance of RGC axons to the optic tectum, and the capacity of retinal neurons and their axons to regenerate (*Adler, 2000*; *Mey and Thanos, 2000*; *Thanos and Mey, 2001*; *Wilken and Reh, 2016*; *Wisely et al., 2017*). To complement and facilitate these studies, we used scRNA-seq to profile cells from the chick retina. From ~40,000 single-cell transcriptomes, we identified cells of all six classes named above (PR, HC, BC, AC, RGC, and glia) and used unsupervised methods to divide them into ~150 clusters. We show that 136 of the groups represent putative cell types, with others corresponding to developmental intermediates. We then devised a method for CRISPR-based somatic cell integration of fluorescent reporters into genes shown by scRNA-seq to be expressed by specific types. Using this technique along with other histological methods, we matched molecular profiles to morphology for many neuronal types. We also found a positional signature in Müller glia, with distinct expression patterns based on their location along the anterior-posterior, dorsal-ventral, and central-peripheral retinal axes. Finally, we compared the cell classes and types of chick retina with those of three mammalian species – mouse, macaque, and human – demonstrating conserved molecular features of all classes and some types, along with multiple differences between chick and mammals. Together, our results provide new insights into retinal structure and evolution, as well as a foundation for anatomical, physiological, and developmental studies of avian retina.

## Results

### Profiling chick retinal cells

All known chick retinal cell types are born by E14, retinal structure is relatively mature by E18, and birds are visually competent at hatching (E21; *Figure 1C*; *Hamburger and Hamilton, 1951*; *Prada et al., 1991*; *Cepko et al., 1996*; *Mey and Thanos, 2000*; *Yamagata and Sanes, 1995a*; *Yamagata and Sanes, 1995b*; *Drenhaus et al., 2003*). We used a droplet-based method (*Zheng et al., 2017*) to obtain 30,022 high-quality single-cell transcriptomes from embryonic day 18 (E18) chick retina (*Figure 1—figure supplement 1A*). We assigned cells to classes based on expression of previously established markers, using methods described in *Peng et al., 2019* and *Yan et al., 2020a*. We identified five neuronal classes (PRs, HCs, BCs, ACs, and RGCs) as well as two glial types, Müller glia and oligodendrocytes (*Figure 1D,E,F*).

Of the E18 retinal cells profiled, 620 were RGCs (*Figure 1F*). This fraction (~2%) was similar to that observed in other species (*Macosko et al., 2015*; *Peng et al., 2019*) but was insufficient for extensive classification of what we anticipated would be a highly heterogeneous class. We therefore used magnetic beads coated with antibodies to the pan-RGC cell surface marker Thy1 (*French and Jeffrey, 1986*; *Yamagata et al., 2002*) to enrich RGCs from E16 retina before scRNA-seq. We obtained 9159 single-cell transcriptomes, of which 8107 (89%) were RGCs (*Figure 1G*; *Figure 1—figure supplement 1B*) that were used for subsequent analysis.

We combined single-cell transcriptomes for all cells except RGCs from the E18 dataset and for all and only RGCs from the E16 dataset to generate a cell atlas. To resolve cell types within these classes, we reclustered each one separately, and obtained a total of 150 clusters (*Figure 1H*). Each presumably corresponded to a cell type, a small group of closely related types, a positional variant or, since we profiled embryonic cells, a developmental intermediate. Quality metrics for each clusters are shown in *Supplementary file 3*.

As one approach to separating definitive types from developmental intermediates, we collected cells from E12 retina, obtaining single-cell transcriptomes from PRs, HCs, BCs, RGCs, and MGs as well as several putative precursor populations (*Figure 1I*; *Figure 1—figure supplement 1C–E*). These cells were not used to generate the atlas, but rather to test developmental hypotheses.

## CRISPR-based cell type characterization

Generation of a retinal cell atlas for mouse benefited from prior knowledge, numerous well-characterized antibodies, and many transgenic lines that express a reporter in one or a few types (*Macosko et al., 2015*; *Shekhar et al., 2016*; *Tran et al., 2019*; *Yan et al., 2020a*). Lacking these advantages for chick, we tested methods for inserting fluorescent reporters into genes shown by scRNA-seq to be expressed by specific types, thereby allowing us to visualize those cells' morphology. In a method called SLENDR, guide RNAs, and plasmids encoding Cas9 and a reporter are delivered to somatic cells to insert the reporter into a chromosomal site determined by the sequence of the guide RNAs and homologous sequences appended to the reporter (*Mikuni et al., 2016*). We were unsuccessful in applying this method to chick retina, and therefore modified it. We used in ovo electroporation to deliver a mixture consisting of guide RNAs, Cas9 protein, and a single-strand DNA containing a reporter sequence flanked by ~70 bases gene-specific homology arms (*Gurumurthy et al., 2019*; *Figure 2A*). We co-electroporated this mixture with piggyBac transposon reporter/transposase constructs encoding a spectrally distinct reporter to monitor the site and efficiency of electroporation, and optimized reagents to enhance homologous recombination (see Materials and methods). We call the method eCHIKIN for *e*lectroporation- and *C*RISPR-mediated *H*omology-*I*nstructed *K*nock-*IN*.

To test eCHIKIN, we targeted genes expressed in most cells of three retinal classes: *VSX2* (Chx10) in BCs, *TFAP2A* in ACs and *RBPMS2* in RGCs (*Chen and Cepko, 2000*; *Bassett et al., 2007*; *Piri et al., 2006*). We inserted a nine amino acid hemagglutinin (HA) tag at the N-terminus of VSX2 and RBPMS2 and inserted GFP into the *TFAP2A* locus. In each case, the homologously inserted reporter specifically labeled the expected cell class, as verified by soma position and immunostaining: VSX2-HA-tagged cells were restricted in the outer part of the INL, as appropriate for BCs, and were *VSX2*-positive and *TFAP2A*-negative (*Figure 2B–D*). TFAP2A-GFP cells were present in the inner portion of the INL as appropriate for *TFAP2A*-expressing ACs (*Figure 2E*). In this case, GFP was not fused to the endogenous proteins, so it filled the cytoplasm, revealing dendrites of ACs in the IPL. In addition, the labeled cells still express the endogenous gene (*Figure 2E*), suggesting that only one allele was edited in most cells. Likewise, RBPMS2-HA-tagged cells were confined to the ganglion cell layer, as appropriate for RGCs (*Figure 2F*). In each case, the spectrally distinct co-electroporated reporter, which had its own regulatory elements and integrated non-homologously, was expressed in cells of all layers (*Figure 2B–F*).

To efficiently label cells that expressed target genes at low levels, we inserted Cre recombinase into the *TFAP2A* and *RBPMS2* loci, and co-electroporated with a Cre-dependent fluorescent protein (loxP-STOP-loxP-GFP under the control of strong and ubiquitously expressed CAGS promoter and enhancer). Cell class specificity was similar in these cases to those in which the loci were tagged directly with a reporter (*Figure 2G,H*).

Thus, eCHIKIN provides a means of matching molecular identity to cellular position, morphology and lamination without germline manipulation.

## Photoreceptors

Chicks are tetrachromatic, with cone types that express red opsin (*OPN1LW*), green opsin (*OPN1MSW*), blue opsin (*OPN2SW*), or violet opsin (*OPN1SW*) (*Kram et al., 2010*; *Enright et al., 2015*). In addition to these conventional 'single-cones' (SCs), the retina also contains rods that express rhodopsin (*RHO*) and *OPN1LW*-expressing 'double-cones' (DCs) composed of tightly apposed principal and accessory cells, which together comprise about half of all PRs (*Smith et al., 1985*; *López-López et al., 2008*; *Oishi et al., 1990*). This composition is unlike that of rodent and primate retinas, which are rod-dominated and contain two or three SC types but no DCs.

Reclustering of PRs revealed 12 clusters (numbered in order of descending abundance; *Figure 3A*; *Figure 3—figure supplement 1A*), including one each expressing *RHO*, *OPN1MSW*, *OPN2SW*, and *OPN1SW* (PR clusters 3, 5, 8, and 11), marking them as rods and green, blue, and violet SCs, respectively (*Figure 3B*).

As expected, in situ hybridizations to opsins labeled non-overlapping cells (*Figure 3C* and data not shown). These types were also distinguished by other differentially expressed genes, including *PDE6B*, *PDE6G* (phosphodiesterases), and *MAFA* (a transcription factor) in rods, *SLC27A6* (a fatty acid transporter) in blue and violet SCs and *LOC101750261* (a non-coding RNA) in green SCs.

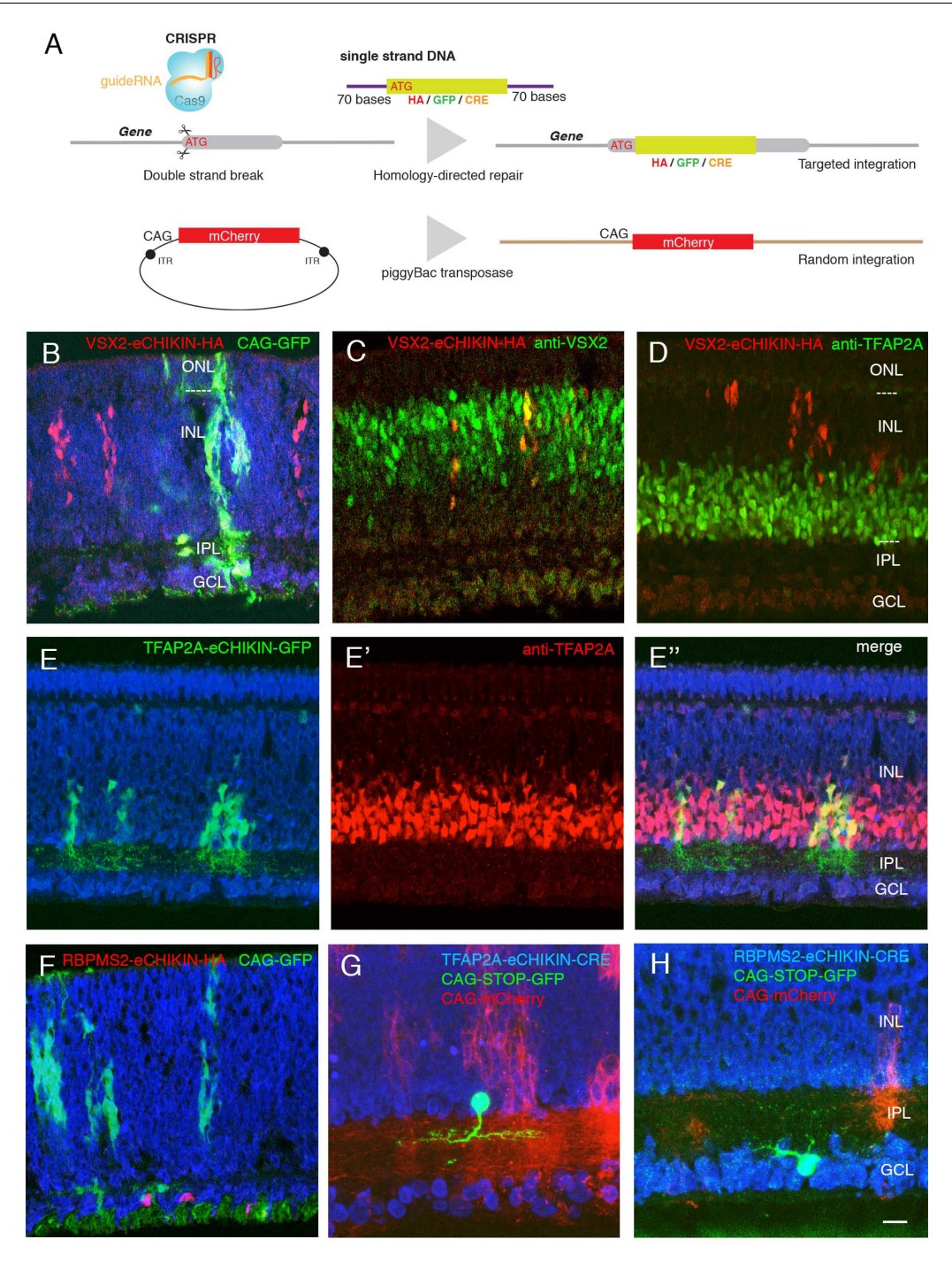

**Figure 2.** Introduction of tags and reporters to specific loci using eCHIKIN. (**A**) The eCHIKIN method. In ovo electroporation of Cas9/guideRNA ribonucleoprotein complexes edits the gene specified by the guide RNA, inserting a sequence encoding HA tag, GFP, or CRE. To identify transfected areas in tissues, a second fluorescent protein (GFP or mCherry) is expressed using the piggyBac transposon system. (**B–D**) Insertion of HA epitope tag into the nuclear protein encoded by *VSX2*. Cells stained by the anti-HA antibody are in the inner (upper) portion of the INL where bipolar cells (labeled with anti-VSX2 in C) are located at E12. CAG-driven GFP is expressed in all the layers. No HA-labeled cells are present in the lower portion of the INL, which contains TFAP2-positive amacrine cells (**D**), and all HA-positive cells are TFAP2-negative. In this and subsequent figures, sections were stained with NeuroTrace (blue) during mounting. (**E, E', E''**) GFP with a termination codon was inserted at the initiation codon of *TFAP2A*, an amacrine cell (AC) marker. In this case, GFP is not fused to TFAP2A protein, resulting in filling cytoplasm including neurites in the IPL at E12. TFAP2A protein is expressed by GFP-expressing cells, all of which are TFAP2A-positive ACs. (Note that the eCHIKIN construct disrupts the *TFAP2A* open-reading frame, so double labeling results from expression of endogenous *TFAP2A* and indicates that only a single *TFAP2A* allele was edited). (**F**) Insertion of HA epitope tag into

*Figure 2 continued on next page*

*Figure 2 continued*

the cytoplasmic protein encoded by RGC-specific *RBPMS2* gene. Labeled cells are in the GCL at E10. (**G**) Insertion of Cre recombinase into the *TFAP2A* gene. The insertion construct was coelectroporated with a CAG- loxP-STOP-loxP-GFP construct, labeling a small number of ACs with GFP at E14. (**H**) Insertion of Cre recombinase into the *RBPMS2* gene. The insertion construct was coelectroporated with CAG- loxP-STOP-loxP-GFP as in F, labeling a small number of RGCs with GFP at E14. Bar in H, 10 μm for B-E,G; 5 μm for F, H.

Expression of *OPN1LW* was detected in four clusters (PR clusters 1, 2, 4, and 7; *Figure 3B*), representing red SCs and DCs. To distinguish among them, we localized cluster seven with *STRA6*, a retinol transporter (*Isken et al., 2008*), and clusters 1, 2, and 4 with *CALB1*. STRA6+ cones were thin, as expected for SCs. In contrast, CALB1+ cells were broader, as expected for DCs (*Figure 3D–F*). We also generated an eCHIKIN probe for *CALB1*, which labeled DCs (*Figure 3G*). Thus, we identify PR7 as red SCs and PR1, 2, and 4 as components of DCs. PR clusters 1, 2, and 4 are quite similar at the gene expression level, so we were unable to distinguish principal from accessory cell components of DCs, or to determine why they formed three clusters rather than the expected two. Their close similarity raises the possibility that besides the gap junctions that connect them (*Smith et al., 1985*) there may be additional ways that enable exchange of mRNAs.

The transcriptomic relationships among these clusters are demonstrated by a dendrogram in *Figure 3B*. There were three major branches, of which two corresponded to rods and cones. Among the cones, SCs are more closely related to each other than to any components of DCs. Among SCs, red and green SCs are each other's closest relatives as are blue and violet SCs.

The third clade of PRs, comprising four clusters, expressed the PR fate determination gene, *PRDM1* (*Katoh et al., 2010*; *Brzezinski et al., 2013*), but did not express opsins, suggesting that they were immature PRs. To relate these clusters to the definitive PRs, we used a supervised classification method, XGBoost (*Chen and Guestrin, 2016*) that predicts the best match among mature types for each putative immature type (*Figure 3H*). PR6 and 9 mapped mostly to the DC types, PR10 mapped to three out of the four SC types, and PR12 mapped almost exclusively mapped to rods. Consistent with this relationship, each immature cluster expressed markers of the corresponding mature PRs – for example, PR6 expressed the DC marker *GRIK3*, and PR10 expressed the rod markers *MAFA* and *PDE6G*.

To further test the idea that the opsin-negative PR-like cells represented immature PRs, we queried our E12 dataset together with the E18 PRs for clustering. E12 cells aligned with both immature and mature types of E18 were present (*Figure 3—figure supplement 1B,C*) but, as expected, the proportions differed with age: immature types comprise 91% of the E12 data set but only 16% of the E18 data set.

Finally, we analyzed retinas from earlier (E12 and E16) and later (E20) stages by in situ hybridization with probes for *ARHGAP18*, and *SLIT1*, expressed by PR6 (immature DCs); *OPN1LW* (mature red SCs and DCs) and *STRA6* (mature SCs). All PRs have been born by E12 (*Prada et al., 1991*), At E12, *ARHGAP18* was expressed by PRs in the ONL, but *OPN1LW* was not detectable. Conversely, *OPN1LW* was expressed at E20 but *ARGHAP18* was not (*Figure 3—figure supplement 2A–D*), suggesting that *ARGHAP18* is transiently expressed in the ONL during development. At E16, consistent with the fact that retinal differentiation proceeds in a central-to-peripheral gradient (*Kahn, 1974*; *Spence and Robson, 1989*; *Prada et al., 1991*; *Bruhn and Cepko, 1996*), *ARHGAP18* and *SLIT1* were expressed by ONL cells selectively in peripheral retina, whereas *OPN1LW* and *STRA6* were by ONL cells in central retina (*Figure 3—figure supplement 2E–Q*). Together, these results strongly suggest that immature and mature PR types co-exist at the same age but in different regions.

## Horizontal cells

Initial histological studies of avian retina distinguished two HC types: Type I, which bears an axon, and Type II, which does not (*Cajal, 1892*; *Mariani and Leure-DuPree, 1977*). Later studies combined immunohistochemical and morphological criteria to propose further subdivisions into three or four types (*Fischer et al., 2007*; *Edqvist et al., 2008*; *Boije et al., 2016*). Clustering of E18 HCs yielded five clusters, HC1-5 (*Figure 4A*, *Figure 4—figure supplement 1A*), all expressing the canonical HC marker *ONECUT3*. *LHX1* and *ISL1* were detected in exclusive populations and other markers distinguished each type (*Figure 4B*). Based on prior immunohistochemical analysis (*Fischer et al.,*

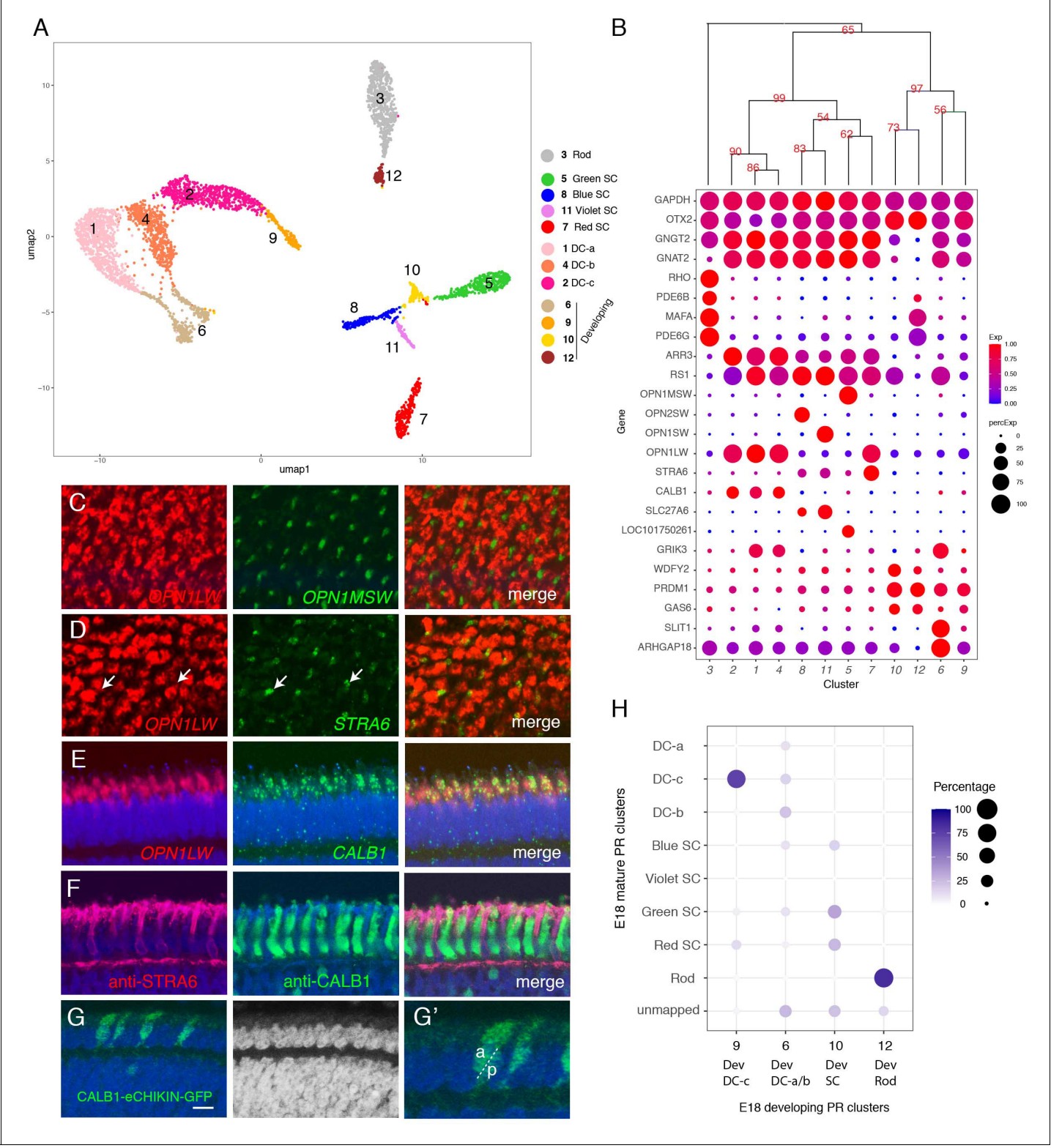

**Figure 3.** Classification and characterization of photoreceptors (PRs). (**A**) Clustering of E18 PRs displayed in UMAP. Identities of each cluster are indicated to the right. (**B**) Dot plots showing expression of selected genes expressed in all or subsets of PRs. In this and subsequent figures, dot size indicates the proportion of cells that express each gene, and color indicates expression level normalized to its max value among clusters. Numbers correspond to clusters in A. Dendrogram above dots shows transcriptional relationships of clusters. In this and subsequent figures, numbers labeled on the tree are p-values computed by multiscale bootstrap resampling, ranging from 0 to 100, higher value indicates higher reliability. (**C–E**) In situ

*Figure 3 continued on next page*

*Figure 3 continued*

hybridization of E16 sections (*en face* in C,D; vertical in E,F) with probes for cluster-specific genes. (C) *OPN1LW* and *OPN1MSW*. (D) *OPN1LW* and *STRA6*. Arrows show coexpression. (E) *OPN1LW* and *CALB1*. (F) Immunostaining of E16 section with antibodies to STRA6 and CALB1. (G) Double cones (DCs) labeled by eCHIKIN-mediated insertion of GFP into the *CALB1* locus. Section is from E17 retina. G' is a high-power picture of a part of G, showing an accessary DC (a) and principal DC (p) based on their position in outer nuclear layer (ONL). Bar in F, 10 μm for C and D; 5 μm for E–G. (H) Relationship between immature and mature PR clusters assessed by XGBoost. Annotation of clusters is indicated in A. Dev, developing.

The online version of this article includes the following figure supplement(s) for figure 3:

**Figure supplement 1.** Frequency distribution of photoreceptors and comparison of E12 and E18 data .

**Figure supplement 2.** Regional distribution of developing photoreceptor cell types.

*2007*; *Edqvist et al., 2008*), we hypothesized that HC1 and HC3 corresponded to Type I and HC2/HC4/HC5 to Type II HCs.

The two type I clusters (HC1 and HC3; *LHX1+ ISL1-*) were closely related, with few genes differentially expressed between them showing large differences (*Figure 4B*). In situ hybridization for two such genes – *IPCEF1* enriched in HC1 and *OXT* enriched in HC3 – confirmed partially overlapping expression (*Figure 4C*). The differences between *IPCEF1+OXT-* and *IPCEF1-OXT+* cells might reflect graded expression in time and/or space. Spatiotemporal analysis of expression similar to that described for PRs above, indicated that *OXT* (enriched in HC3) was expressed throughout the retina at E14, restricted to peripheral at E18, and disappeared at E20 (*Figure 4—figure supplement 1B–G*). In contrast, *IPCEF1* (enriched in HC1) was expressed selectively in central retina at E14 and E18, and strongly throughout retina at E20 (*Figure 4—figure supplement 1H–M*). Consistent with this conclusion, most of the HCs identified in the E12 collection expressed OXT but not IPCEF1 (*Figure 4—figure supplement 1N–P*). Thus, we conclude that HC1 is a mature type and HC3 its immature counterpart.

The *ISL1+LHX1-* types HC2, HC4, and HC5 were distinguished by selective and non-overlapping expression of distinct receptor-type tyrosine kinases, *NTRK1*, *EGFR*, and *LTK*, respectively (*Figure 4B,D–I*). In situ hybridization of *en face* sections demonstrated that these groups form mosaics of HCs, with the density of each mosaic corresponding roughly to the abundance of the type observed by scRNA-seq (*NTRK1>EGFR>LTK*; *Figure 4J–L*). NTRK1 (*TrkA*) is expressed by 'Candelabrum' axon-less HCs (*Edqvist et al., 2008*) which correspond to the abundant HC2.

## Bipolar cells

Using Golgi staining, *Quesada et al., 1988* identified 14 types of BCs in chick based on dendritic morphology, and noted that further subdivisions might be possible if axonal morphology was also considered. Unsupervised clustering of our data identified 22 groups of BCs (*Figure 5A*) ranging in frequency from 1.5 to 10.4% of all BCs (*Figure 5—figure supplement 1A*). All expressed the canonical markers *VSX2* and *OTX2*, and each was distinguished by selective expression of other genes (*Figure 5B*). Immunostaining and in situ hybridization for several of these genes confirmed their expression by BC subsets (*Figure 5—figure supplement 1B–O*).

In mammals, BCs are conventionally divided into two groups, based on whether they respond to illumination with depolarizing (ON) or hyperpolarizing (OFF) responses; BCs innervated by cones can be either ON or OFF, whereas those innervated by rods are all ON type (*Euler et al., 2014*). In mammals, ON BCs are characterized by expression of *TRPM1*, *ISL1*, and *GRM6*; OFF BCs express *GRIK1*; and rod BCs express *PRKCA* in addition to canonical ON markers (*Morgans et al., 2009*; *Morgans et al., 2010*; *Shekhar et al., 2016*). In our dataset, the expression of GRIK1 was mutually exclusive with that of *TRPM1* and *ISL1* in all clusters but one, resulting in 11 likely ON types and 10 likely OFF types (*Figure 5C*). (GRM6, the canonical marker of mammalian ON BCs is missing in the current version of the chick genome, so we were unable to assess its expression.) In situ hybridization for *GRIK1* and *TRPM1* confirmed that these genes are also expressed in a largely non-overlapping pattern in chick retina (*Figure 5—figure supplement 1B,C*; see below). Supporting this assignment, *FEZF2*, which marks some OFF BC types in mice (*Shekhar et al., 2016*) was enriched in 5 OFF types. *PRKCA* was expressed at highest levels in BC19, a putative ON cluster, suggesting that it might represent rod BCs (*Greferath et al., 1990*). BC10, which expresses both ON and OFF markers might have mixed properties, as has been seen in fish (*Yn et al., 2012*). In general, ON BCs

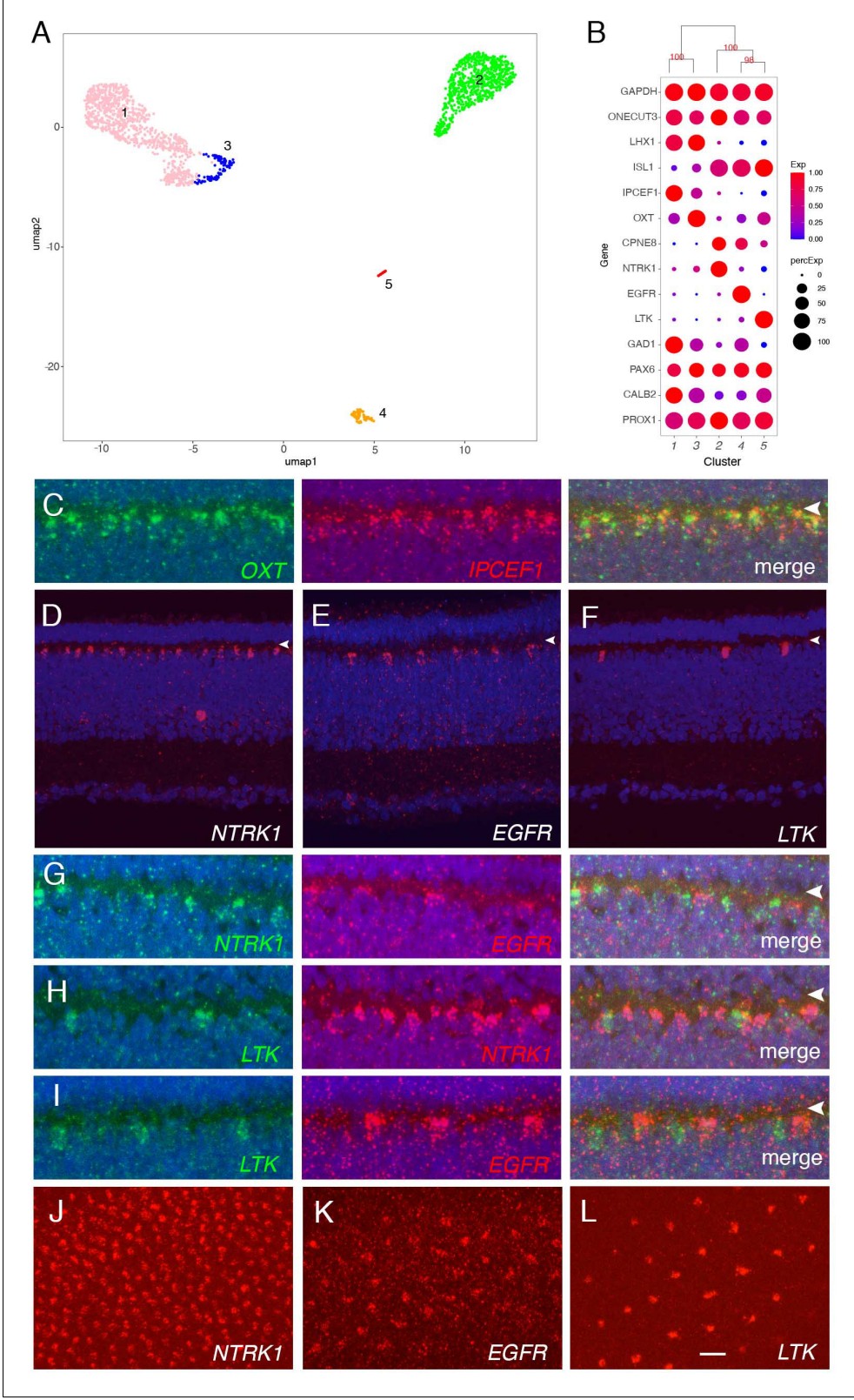

**Figure 4.** Classification and characterization of horizontal cells (HC). (**A**) Clustering of E18 HCs viewed by UMAP. (**B**) Dot plots showing expression of selected genes expressed in all or subsets of HCs. Numbers correspond to clusters in A. Dendrogram above dots shows transcriptional relationships of clusters. (**C–L**) In situ hybridization with indicated probes at E16. C–I are cross-sections; J–L are en face sections. Arrowheads in C–I mark OPL. (**C**)

*Figure 4 continued on next page*

*Figure 4 continued*
Double color in situ hybridization shows coexpression of OXT and IPCEF1. (**D–F**) Expression of NTRK (**D**), EGFR
(**E**), and LTK (**F**) in subsets of HCs. (**G–I**) Double color in situ hybridization for NTKR/ EGFR (**G**), LTK/ NTRK (**H**), and
LTK /EGFR (**I**). (**J–L**) In situ hybridization of E16 en face sections for NTRK (**J**), EGFR (**K**), and LTK (**L**) showing
mosaics of each type. Bar in L, 5 μm for C, G–I; 10 μm for D–F, J–L.
The online version of this article includes the following figure supplement(s) for figure 4:

**Figure supplement 1.** Frequency and regional distributions of horizontal cell types.

were transcriptomically more closely related to other ON BCs than to OFF BC and vice versa
(*Figure 5B,C*), but the segregation was not as strict as in mice or primates, in which ON and OFF
BCs form separate clades (*Shekhar et al., 2016*; *Peng et al., 2019*; *Yan et al., 2020b*).

A key feature of BC types is the IPL sublamina or sublaminae in which their axons arborize. We
used eCHIKIN and immunohistochemistry to assess lamination, adopting the convention of dividing
the IPL into five strata (S1-5). We generated eCHIKIN probes for 5 BC types: BC1 (*ANGPT2*), BC6
(*TPBGL*), BC8 (*RRAD*), BC12 (*IRX3*), and BC15 (*SLC6A4*). As shown in *Figure 5D–H*, BC1 axons termi-
nated in S3, BC6 and BC8 axons in S3, and BC15 in S4; BC12 axons were bistratified with termini in
S1 and S2. Antibodies to TPBGL (BC6) and SLC6A4 (BC15) stained S3 and S4, respectively, consis-
tent with these assignments (*Figure 5I,J*); anti-STRA6 (BC3,9) and anti-ERBB4 (BC12) stained S3-4
and S1-2, respectively (*Figure 5—figure supplement 1Q,R*).

In mammals, there is a relationship between the axonal lamination of BCs and their response
properties, with ON cone, OFF cone and rod BC axons arborizing in the order OFF cone > ON cone
> rod proceeding from the INL to the GCL. In general, the lamination patterns observed with eCHI-
KIN and immunohistochemistry followed this rule with three putative OFF cone types (BC6, 8, and
12) laminating in S1 and S2, two putative ON cone types (BC3 and 18) laminating in S3 and S4, and
putative rod bipolars (BC19; PRKCA+) laminating in S5 (*Figure 5K*). The sole exception was BC15,
which was *GRIK+* but laminated in S4.

Finally, we asked whether the positions of BC somata in the INL were related to the positions of
their axons in the IPL. Somata of putative OFF (*GRIK1+*) and ON (*TRPM1+*) BCs were situated in the
outer and inner portions of the INL, respectively, corresponding to the positions of their axon termi-
nals (*Figure 5—figure supplement 1B,C*). To assess correspondence for individual types, we deter-
mined somata position by in situ hybridization and axon position as described above. For most
types, somata position in INL was correlated with terminal arborization position in IPL (quantified in
*Figure 5—figure supplement 1P*, and summarized in *Figure 5L*). Interestingly, the somata of BC10,
the putative ON-OFF type (see above), populate the interface between ON and OFF regions in the
INL as revealed by SOX5 immunostaining (*Figure 5—figure supplement 1J,P* and *Figure 5L*). This
correlation has not, to our knowledge, been observed for mammalian BCs.

## Amacrine cells

ACs are a diverse class of interneurons, most of which form inhibitory (GABAergic or glycinergic)
synapses on BCs, RGCs, and other ACs. In mammals, ACs are the most heterogeneous retinal class
(*Yan et al., 2020a*; *Yan et al., 2020b*; *Peng et al., 2019*). Similarly, ACs formed the most heteroge-
neous class in chicks, with 59 putative types (*Figures 1G* and *6A*), ranging in frequency from 0.4 to
7.2% of all ACs (*Figure 6—figure supplement 1A*). All expressed *SLC32A1*, a transporter that loads
both GABA and glycine into synaptic vesicles, and each type expressed either GABAergic markers
(GABA transporter, *SLC6A1*, and the GABA synthetic enzymes, *GAD1* and *GAD2*, 40 clusters) or gly-
cinergic markers (glycine transporter 1, *SCL6A9*, 19 clusters) (*Figure 6B*). AC somata were present in
both the INL and the GCL. Expression of two broadly expressed member of the AP2 family of tran-
scription factors, *TFAP2A* and *TFAP2B*, distinguished these populations: *TFAP2A* expression was
restricted to ACs in the INL, while *TFAP2B* was expressed by ACs in both locations (*Figure 6B,D,E*).

We identified markers selectively expressed by one or a few types (*Figure 6C*). Many were neuro-
peptides, consistent with classical immunohistochemical studies (*Karten and Brecha, 1983*;
*Brecha et al., 1984*; *Karten et al., 1990*); they included *TAC1* (substance P), *NPY* (neuropeptide Y),
*PENK* (enkephalin), *NTS* (neurotensin), and *NMB* (neuromedin-B). We used in situ hybridization or
immunohistochemistry to validate selective expression of several markers in AC subsets (e.g. *NMB* in

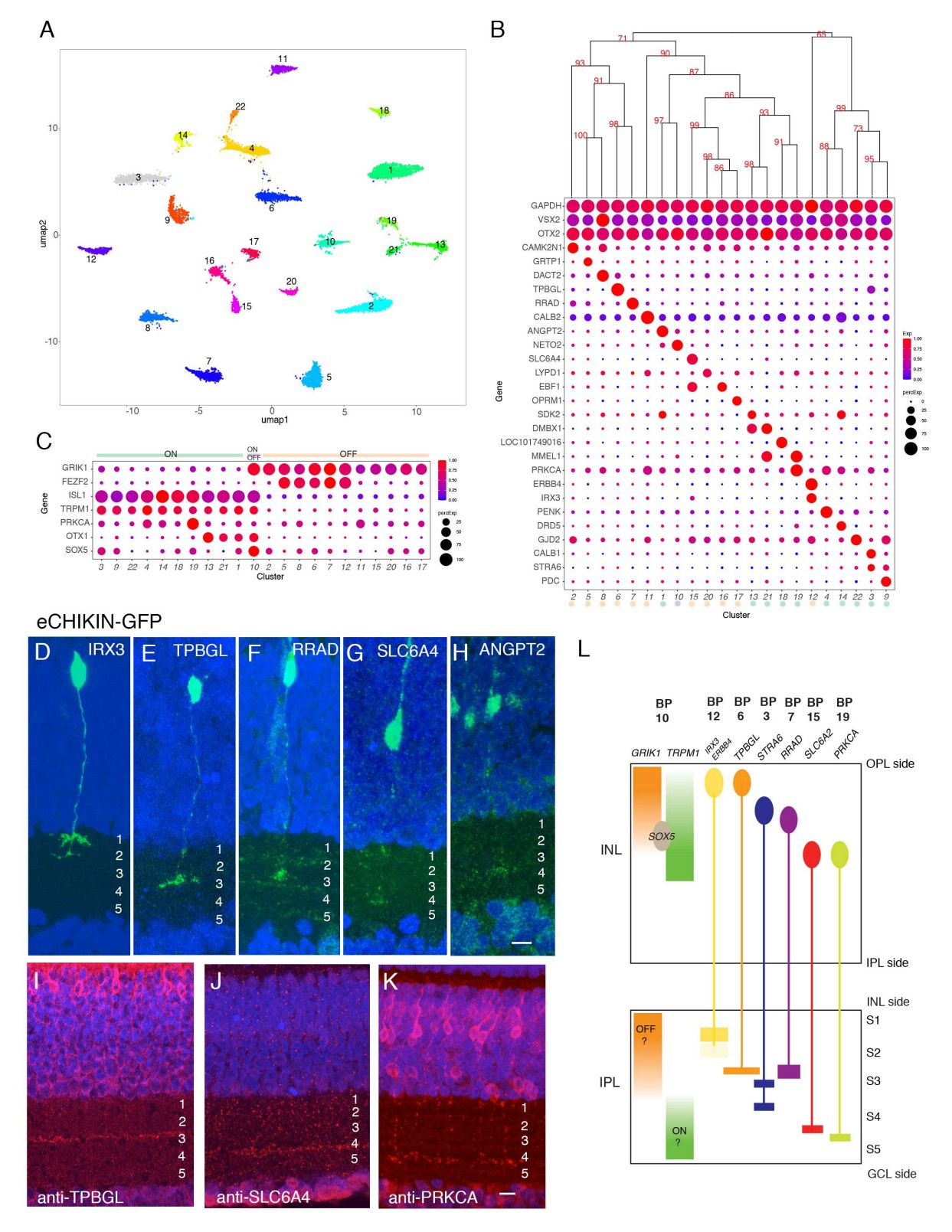

**Figure 5.** Classification and characterization of bipolar cells (BCs). (**A**) Clustering of E18 BCs visualized by UMAP. (**B**) Dot plots showing expression of selected genes expressed in all or subsets of BCs. Numbers correspond to clusters in A. Dendrogram above dots shows transcriptional relationships of clusters. Putative ON and OFF types, based on markers in C, are indicated by color. (**C**) Expression of genes selectively expressed by ON (*TRPM1, ISL1*), OFF (*GRIK1, FEZF1*) and rod (*PRKCA*) BCs in rodents. *OTX1* and *SOX5* are uniquely expressed in putative ON clusters. (**D–H**) eCHIKIN-mediated

*Figure 5 continued on next page*

Figure 5 continued

labeling of cells expressing *IRX3* (BC12, (D)), *TPBGL* (BC6, (E)), *RRAD* (BC7, (F)), *SLC6A4* (BC15, (G)), and *ANGPT2* (BC1, (H)). E14 sections at E14 were stained with anti-GFP. Each lamination was confirmed in 3–10 cases. Bar 10 µm. (I–K) Immunostaining with anti-TBBGL (I), anti-SLC6A4 (J), and anti-PRKCA (K). Bar 10 µm. (L) Summary of BC soma positions in INL and terminal positions in IPL.

The online version of this article includes the following figure supplement(s) for figure 5:

**Figure supplement 1.** Frequency distribution and laminar position of bipolar cell types.

AC17; *CHODL* in AC40; *NPY* in AC52; *NTS* in AC31 and 58; *PENK* highest in AC31, 34, and 42; and *MAFA* in AC35 and 58; *Figure 6C* and *Figure 6—figure supplement 2B–E*). Double label studies distinguished sets of *NTS+PENK+* and *NTS+MAFA+* double positive ACs, corresponding to AC31 and AC58, respectively (*Figure 6—figure supplement 1F–H*). An eCHIKIN probe for *NTS* also labeled two AC types, a more abundant one with broad processes that ramify in S1, S3, and S4, morphologically reminiscent of pigeon *NTS+* ACs (*Brecha et al., 1984*), and a less abundant one with arbors in S5 (*Figure 6—figure supplement 2J,K*).

We further investigated two AC types that have been studied extensively in mammals. One is the starburst amacrine cell (SAC), the only retinal cholinergic cell type. It comprises cohorts in both layers, called ON (somata in the GCL, with dendrites in S4) and OFF (somata in the INL, with dendrites in S2; *Figure 6F*; *Millar et al., 1987*). Two AC clusters, AC7 and AC25, expressed the cholinergic markers choline acetyltransferase (*CHAT*), choline transporter (*SLC5A7*), and vesicular acetylcholine transporter (*SLC18A3*), indicating their identity as SACs. We assign AC7 and AC25 to ON and OFF SACs, respectively, because AC7 expressed *FEZF1*, a transient marker of developing ON starburst marker in mouse, whereas AC25 expressed *TENM3* and *ZFHX3*, transient OFF markers SAC markers in mouse (*Peng et al., 2020*; *Figure 6—figure supplement 2A*).

The other is an unusual excitatory AC, the VG3 amacrine, which expresses the vesicular glutamate transporter 3 (*SLC17A8*, VGlut3). Two closely related chick AC clusters, AC37 and AC39 were *SLC17A8*-positive, with levels higher in AC37 than AC39 (*Figure 6C* and *Figure 6—figure supplement 2B*). AC37 also expressed the recognition molecule Sidekick 2 (*SDK2*; *Yamagata et al., 2002*), which is also a selective marker of mouse VG3 ACs (*Krishnaswamy et al., 2015*; *Yamagata and Sanes, 2018*), suggesting that AC37 is the authentic chick VG3 AC (*Figure 6—figure supplement 2D*). *SDK1*, the homolog of *SDK2*, is expressed at highest levels in AC38, but not in chick VG3 AC (*Figure 6—figure supplement 2B,C*). We generated an eCHICK probe from *ERBB4*, which is expressed at highest levels in AC37 and AC38 among ACs, and used it to mark these cells. They stratified in S2 and S4 (*Figure 6—figure supplement 2E*), consistent with Sdk1 and Sdk2 localization reported previously (*Yamagata et al., 2002*) but distinct from mouse VG3 ACs, which stratify in S3.

## RGCs

From 8107 single RGC transcriptomes, we resolved 41 clusters (*Figure 7A*) ranging in abundance from 0.6 to 5.1% of all RGCs (*Figure 7—figure supplement 1A*). Like RGCs in mammals, all expressed the canonical RGC markers *RBPMS* and *THY1*, as well as one or more of the Brn3 (*POU4F*) transcription factors and the *RBPMS* homolog, *RBPMS2* (*Figure 7B*). Each cluster could be specified by selective expression of one or, in some cases, a few genes (*Figure 7B,C*; *Figure 7—figure supplement 1B–I*). They include genes that are expressed by subsets of RGCs in mammals such as *SATB1* and *SATB2* (*Peng et al., 2019*; *Figure 7D,E*). Among the 41 putative RGC types, cluster 11 selectively expressed *OPN4.1*, a homolog gene to the defining marker of mammalian intrinsically photosensitive RGCs (ipRGCs), as well as the transcription factor *EOMES* (*Tbr2*) (*Figure 7B*; *Figure 7—figure supplement 1I*), which is expressed in all but not only ipRGCs (*Chaurasia et al., 2005*; *Mao et al., 2014*), indicating that GC11 is a chick ipRGC type. We used eCHIKIN to reveal morphologies of three RGC types: GC23 (*TFAP2D*), with dendrites in S5; GC18 (*MC5R*), with dendrites in S5; and GC15 and/or 13 (*ETV1*) with dendrites in S4; (*Figure 7F–H*). GC15 selectively expresses the recognition molecule *SDK1*, which we have shown to be concentrated in S4 (*Yamagata et al., 2002*; *Figure 7—figure supplement 2A,B*).

The main central target of chick RGCs is the optic tectum. Within the tectum axons of distinct RGC populations terminate in one of five retinorecipient laminae, called B, C, D upper, D lower and F (*Yamagata and Sanes, 1995a*; *Yamagata and Sanes, 1995b*; *Yamagata et al., 2006*). Based on

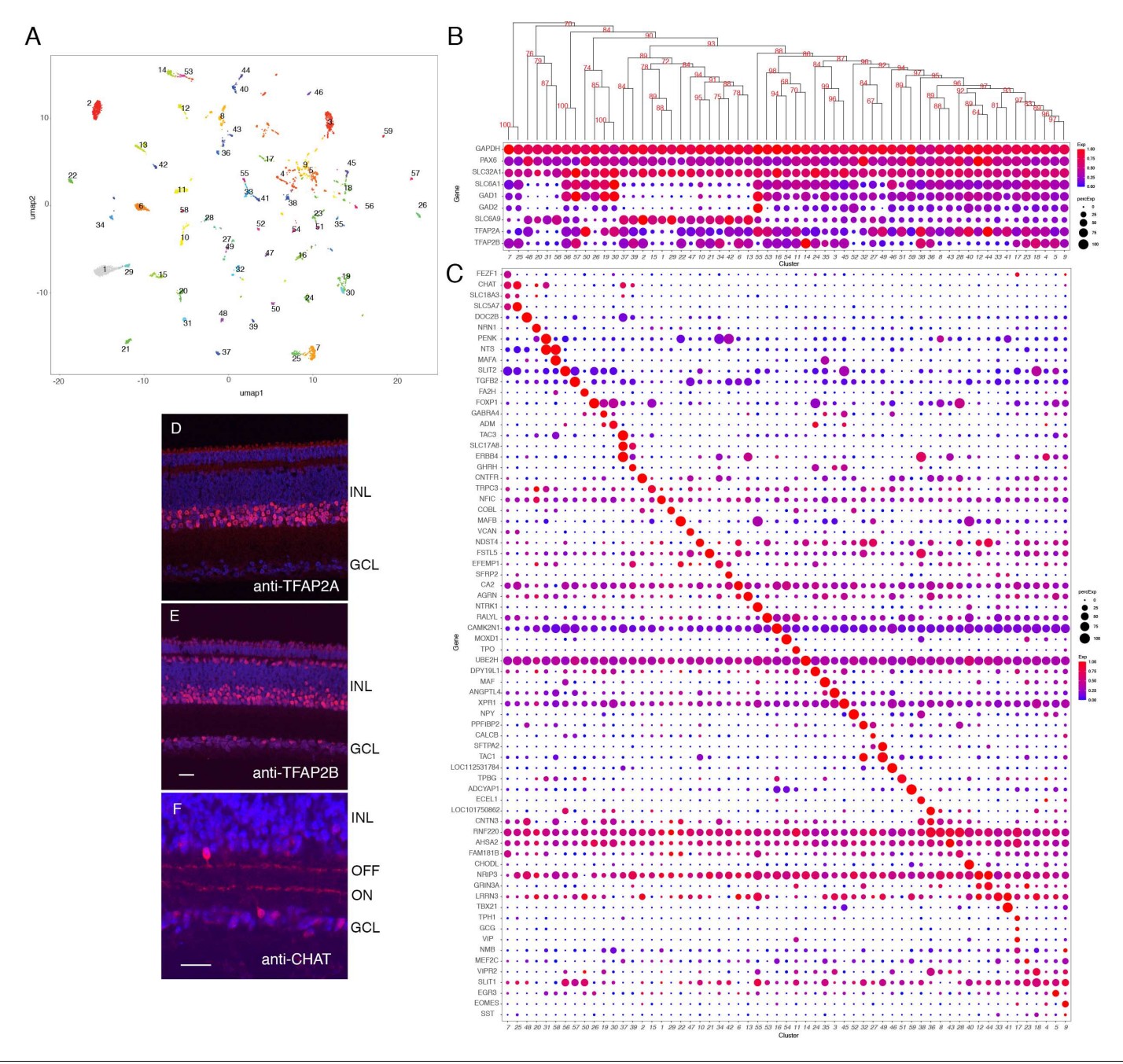

**Figure 6.** Classification and characterization of amacrine cells (ACs). (**A**) Clustering of E18 ACs using UMAP. (**B**) Dot plots showing expression of the housekeeping gene, *GAPDH*; pan-AC genes *PAX6* and *SLC32A1*; genes diagnostic of GABAergic ACs (*SLC6A1*, *GAD1*, *GAD2*) and glycinergic ACs (*SLC6A9*); and *TFAP2* isoforms, *TFAP2A* and *TFAP2B*. Numbers correspond to clusters in A. Dendrogram above dots shows transcriptional relationships of clusters. (**C**) Genes expressed by subsets of ACs. (**D,E**) Immunostaining of E16 retina for TFAP2A (**D**) and TFAP2B (**E**). TFAP2A is expressed by multiple amacrine types in INL but not in GCL. (**F**) Immunostaining of E16 retina for CHAT, which is expressed by ON and OFF starburst ACs. Bar in E is 10 μm for D,E. Bar in F is 10 μm for F.

The online version of this article includes the following figure supplement(s) for figure 6:

**Figure supplement 1.** Frequency distribution and morphological analysis of amacrine cell types.

**Figure supplement 2.** Molecular and morphological analysis of key amacrine cell types.

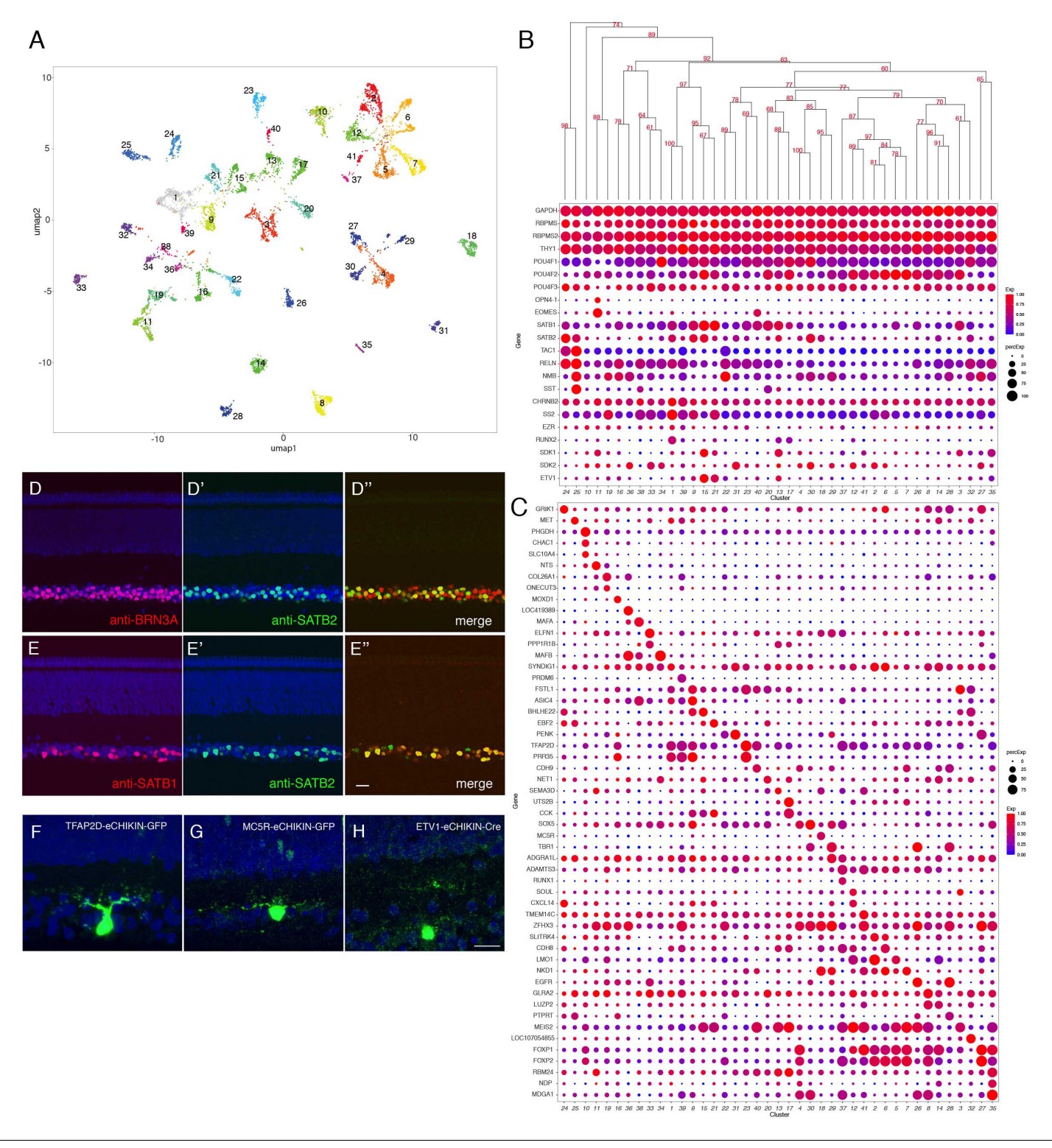

**Figure 7.** Classification and characterization of RGCs. (A) Clustering of E16 RGCs using UMAP. (B) Dot plots showing expression of selected genes expressed in all or subsets of RGCs. Numbers correspond to clusters in A. Dendrogram above dots shows transcriptional relationships of clusters. (C) Dot plots showing expression of genes selectively expressed in RGC clusters. (D, E) Immunostaining of E16 retina for SATB2 and BRN3A/POU4F1 (D) and for SATB1 and SATB2 (E). (F–H) RGCs labeled by eCHIKIN-mediated GFP insertion in *TFAP2D* (GC23; F), *MC5R* (GC18; G), and *ETV1* (GC15; H). Bars in E and H, 10 µm.

*Figure 7 continued on next page*

*Figure 7 continued*

The online version of this article includes the following figure supplement(s) for figure 7:

**Figure supplement 1.** Frequency distribution and morphological analysis of retinal ganglion cell types.

**Figure supplement 2.** Markers expressed and co-expressed by key retinal ganglion cell types.

expression of markers identified in previous studies, we conclude that GC25 targets tectal lamina B (*TAC1*[substance P]+, *NMB* [neuromedin B]+, *SST* [somatostatin I]+, and *RELN* [reelin]+) and GC1 targets tectal lamina F (*CHRNB2* [neuronal acetylcholine receptor beta2 subunit]+, *SS2* [somatostatin II], and EZR [ezrin]+) (*Yamagata et al., 2006*). GC1 is characterized by expression of a transcription factor *RUNX2* (*Figure 7B*, *Figure 7—figure supplement 2C–E*).

## Müller glia

Müller glia, the major retinal glial type, has generally been viewed as a homogeneous population (see Discussion). However, we distinguished five clusters in the single-cell dataset of E18 chick retina (*Figure 8A and B*). In examining genes differentially expressed among these clusters (*Figure 8C*), we noted three that had been shown to exhibit topographically biased expression at early stages of retinal development, when most cells are still mitotically active: *CHRDL1* (ventropin), expressed in ventral retina was enriched in cluster MG1 (*Sakuta et al., 2001*); and *EPHA3* and *FOXD1* (BF2), expressed in temporal retina, were enriched in MG5 (*Cheng et al., 1995*; *Yuasa et al., 1996*; *Yamagata et al., 1999*). This suggested that positional differences might underlie Müller glia heterogeneity.

We used in situ hybridization to test this idea. We documented selective expression of *CHRDL1* (MG1) in ventral retina, *WIF1* (MG2) in dorsal retina, *FOXI2* (MG5) in temporal retina, and *FOXG1* (BF1) in nasal retina (*Figure 8D–M*). *FOXG1* was detected in a subset of cells, that failed to form a single cluster but presumably comprise nasal cells (*Figure 8C*). Finally, in situ hybridization for genes selectively expressed in MG3 (*PSCA*) and MG4 (*TMEM123*) indicated that cells in these clusters were associated with central and peripheral retina, respectively (*Figure 8N–Q*). Expression of all markers was graded with position. Co-staining for glutamine synthetase, an MG marker (*Linser and Moscona, 1979*), showed that all these genes were selectively expressed in MG (*Figure 8—figure supplement 1*). Quantitative analysis confirmed both the distinct expression patterns of genes expressed by each cluster as well as the partial overlap expected for graded expression and the dual identity of, for example, cells in the ventral quadrant of central retina (*Figure 8—figure supplement 2A*). Together, these results reveal a striking positional map of gene expression in MG (*Figure 8R*).

In light of the central-to-peripheral developmental gradient documented above we wondered whether MG3 (central) and MG4 (peripheral) represented authentic positional differences or different developmental stages. To distinguish these alternatives, we performed two additional analyses. First, we queried the E12 dataset. Markers of dorsal, ventral, nasal, and temporal retina were selectively expressed, but the peripheral marker at E18, *TMEM123*, was broadly expressed at E12, suggesting that it is a marker for an early developmental stage (*Figure 8—figure supplement 2A,B*; *Figure 8—figure supplement 3A*). *PSCA* was barely detectable expressed at this stage, but FGF8, known to mark a small central retinal region (*da Silva and Cepko, 2017*) was selectively expressed by a restricted group of cells (*Figure 8—figure supplement 3A*). Second, we used in situ hybridization to assess the distribution of key genes at E12 and E20. As development proceeds, *PSCA* and *FGF8* are expressed in nested domains, suggesting further distinctions along the central-peripheral axis (*Figure 8—figure supplement 3B*). At later stages, *FGF8* expression declines and *TMEM123* is progressively restricted to peripheral regions (summarized in *Figure 8—figure supplement 3C*). Thus, MG3 represents a positionally restricted cell group, whereas MG4 may largely represente immature MGs. In summary, our analysis revealed a positional basis for the transcriptomic heterogeneity of MG.

## Oligodendrocytes

Reclustering of oligodendrocytes revealed five clusters (*Figure 9A*; *Figure 9—figure supplement 1A*). All expressed the oligodendrocyte marker *OLIG2* (*Zhou et al., 2001*), but they exhibited

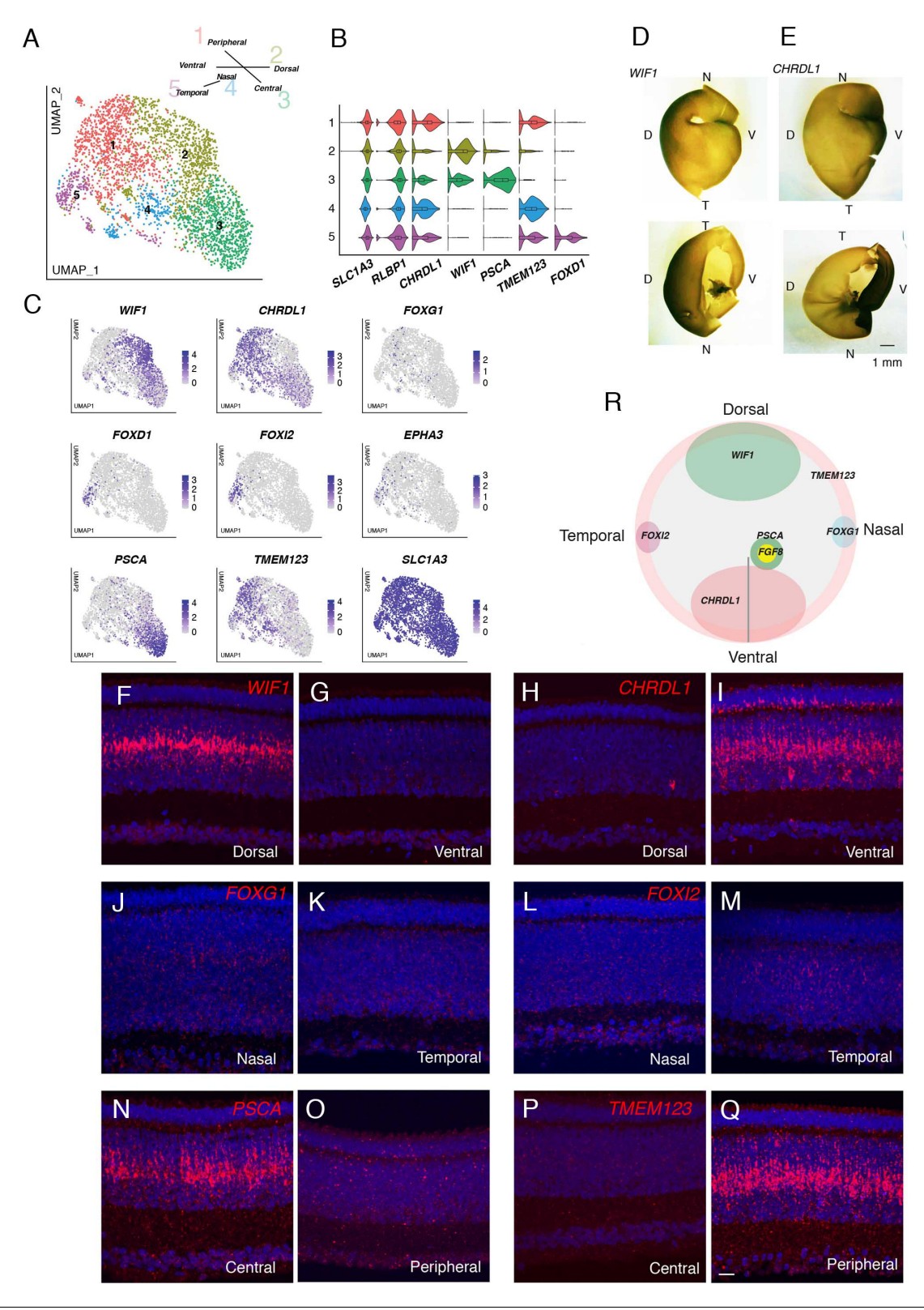

**Figure 8.** Transcriptomic map of topographic position in Müller glia. (A) Clustering of E18 MGs using UMAP. Inset shows relationship between clusters and retinal position. (B,C) Violin (B) and feature (C) plots of genes differentially expressed among MG clusters. B also shows that pan-MG genes *SLC1A3* and *RLBP1* are expressed at similar levels among clusters. (D, E) In situ hybridization for *WIF1* (D) and *CHRDL1*(E) on whole mounts at E13 photographed from the posterior (top panels) or anterior (bottom panels). The black structure at the ventral edge in the bottom panels is the

*Figure 8 continued on next page*

*Figure 8 continued*

intrinsically pigmented pecten oculi. Bar, 1 mm. (F–Q) In situ hybridization on sections from indicated retinal regions to show position-selective of genes from C in Müller glia. F-I *WIF1* and *CHRDL1* on E16 dorsal and ventral sections. *CHRDL1* is also in a subset of amacrine cells throughout the retina. (J–M) *FOXG1* and *FOXI2* on E14 nasal and temporal sections. (N–Q) *PSCA* and *TMEM123* on E16 central and peripheral sections. Bar, 10 μm. (R) Summary of position-dependent expression of genes in Müller glia at E16, based on images such as those in D-Q.

The online version of this article includes the following figure supplement(s) for figure 8:

**Figure supplement 1.** Regional distributions of Müller glial cell positional variants.

**Figure supplement 2.** Co-expression of positional markers in Müller glia.

**Figure supplement 3.** Developmental trajectories of Müller glial cell variants.

differential expression of other known markers of developing and mature oligodendrocytes (reviewed in *Goldman and Kuypers, 2015*), suggesting that they represented different developmental stages. Pseudotime analysis arranged the clusters in the order: OL5, OL3, OL1, OL2, and OL4, and selectively expressed markers supported this order. For example, *HES1*, a marker of oligodendrocyte precursors is expressed in the order OL5>OL3,OL1>OL2,OL4; myelin components such as *PLP1* and *MBP* are expressed at highest levels in OL2 and OL4; and *CLDN11*, a component of tight junctions formed in compacted myelin, is selectively expressed by OL4 (*Figure 9B* and *Figure 9—figure supplement 1B*). Several other genes exhibited similar differential expression, making them candidate markers of successive stages in oligodendrocyte differentiation (*Figure 9—figure supplement 1C*).

We used in situ hybridization to localize oligodendrocytes in retina, using probes for *PLP1* and *BCAS1* (OL2 and OL4) and *PDGFRA* (OL1 and OL3). *PLP1* and *BCAS1* were co-expressed, whereas *PDGFRA*+ cells formed a distinct population (*Figure 9B*; *Figure 9—figure supplement 1B,C*). Both populations were confined to the GCL. *BCAS1+PLP1*+ cells were more abundant in central retina, near the optic disc, than in the periphery, whereas *PDGFRA*+ cells were more abundant peripherally than centrally (*Figure 9C–L*). These patterns are consistent with the idea that oligodendrocyte precursors enter from the optic nerve and migrate peripherally, with progeny maturing in a central-to-peripheral gradient (*Ono et al., 1997*; *Fischer et al., 2010*).

## Comparison of avian and mammalian retinal cell classes

As noted in the Introduction, the basic retinal plan, including the structure and placement of its main cell classes, is conserved among vertebrates. We asked whether this morphological conservation is accompanied by transcriptomic conservation. To this end, we combined data from the E18/E16 chick atlas with those from our previously published retinal atlases of mouse (*Macosko et al., 2015*; *Shekhar et al., 2016*; *Tran et al., 2019*; *Yan et al., 2020a*), macaque (*Peng et al., 2019*), and human (*Yan et al., 2020b*), and submitted the entire group for clustering. This procedure generated nine clusters, all of which contained cells from all species (*Figure 10A,B*). Each cluster could be identified with high confidence by expression of class-specific markers (*Figure 10C*) and by reference to prior assignments made when each species was analyzed individually (*Figure 10D*). Six of the clusters corresponded to major retinal cell classes: rods, cones, HCs, BCs, RGCs, and MG. The other three clusters were composed of ACs distinguished by neurotransmitter type: GABAergic, glycinergic, and cholinergic+GABAergic (SACs); these distinctions are discussed in the next section.

Transcriptomic relationship among classes are shown in *Figure 10E*. The highest level split is between MG and neurons; the second separates PRs from other neurons; the third separates HCs from other interneurons and RGCs; and the last separates GABAergic from glycinergic ACs. These patterns are consistent with structural and functional data, in that glia and neurons play largely non-overlapping roles, PRs are highly specialized neuron-like cells, HCs are unique among interneurons generally, and the GABAergic and glycinergic ACs are inhibitory interneurons with similar functions and patterns of connectivity. The greater similarly of RGCs to ACs than to BCs is also not unexpected in that ACs and RGCs use conventional neurotransmission mechanisms, whereas BC bear ribbon synapses that are shared with PRs but few other neuronal types. On the other hand, the distance placement of SACs is unexpected. All four species obey the same rules of similarity.

In contrast, relationships among species vary by class. Humans and macaques are each other's closest relatives in only four of nine classes, and mice rather than chicks are outliers in seven of nine

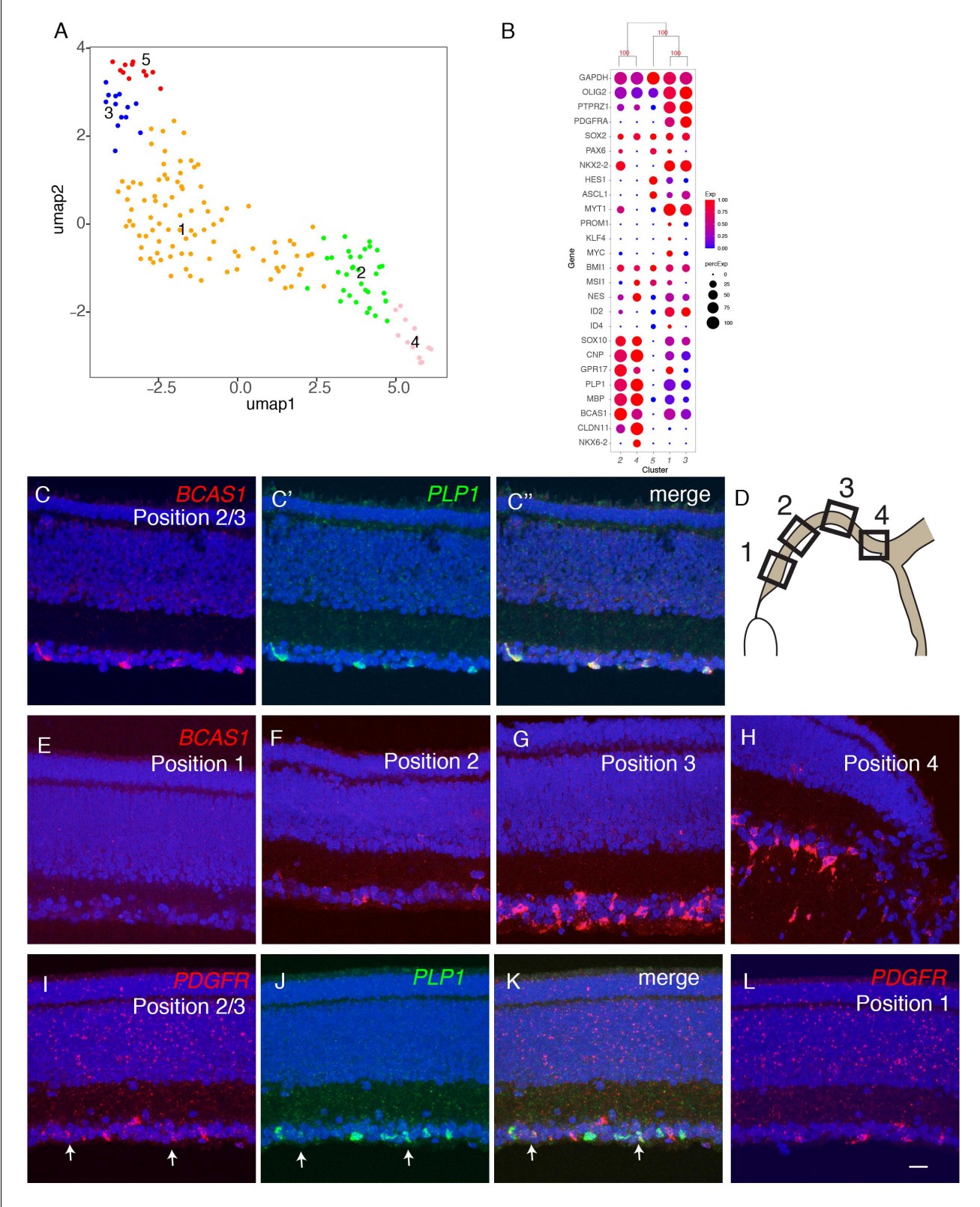

**Figure 9.** Developmental trajectory of oligodendrocytes. (A) Clustering of E18 oligodendrocytes viewed in UMAP. (B) Dot plots showing expression of selected genes expressed in all or subsets of oligodendrocytes. Numbers correspond to clusters in A. (C-K) Double-label in situ hybridization showing that *BCAS1* and *PLP1* are co-expressed in the ganglion cell layer (C) while *PDGFR* and *PLP1* exhibit nonoverlapping expression (I-K). (E-L) Graded distribution of *BCAS1*+ and PDGFR+ oligodendrocytes along the central-to-peripheral axis at E16. Positions of sections (E–L) are shown in D.

*Figure 9 continued on next page*

*Figure 9 continued*

The online version of this article includes the following figure supplement(s) for figure 9:

**Figure supplement 1.** Frequency distribution and molecular analysis of oligodendrocyte variants.

cases; both of these patterns are unexpected from phylogenetic considerations. The differences are small, however, and may result from technical considerations: the highly variable genes used for clustering may not be sufficient to distinguish cells within any individual class, as the high dimensional space is saturated by genes for other classes. Indeed, the relationship among species differ when each class is clustered separately (*Figure 11*, see below). Nonetheless, when all cells are combined and compared by species, retinal cells from chick and mouse are transcriptomically more similar to each other than either is to primates (*Figure 10F*). We have no explanation for this seeming anomaly. One possibility is that chick and mouse retina are both more complex in terms of numbers of cell types than either human or macaque retina (*Table 1*).

## Comparison of avian and mammalian retinal cell types

Finally, we examined the conservation of cell types within classes for the four species, using two methods: building a dendrogram based on similarity matrix of all types from all species (*Figure 11* and *Figure 11—figure supplements 1* and *2*) or pooling cells within each class from all species and submitting the combined data for clustering (shown for ACs and RGCs in *Figure 11—figure supplements 1* and *2*). Only mature chick types were used for this analysis. Key findings by class are as follows:

Cones (*Figure 11A*): Mammalian M/L and S cones form separate clades. (Primate M and L cones differ transcriptomically only at the opsin locus [*Peng et al., 2019*] and are combined here. Most mouse cones are M type, so we had insufficient power to analyze mouse S cones as a separate type). Chick cones are outliers, but red (L) SCs are most closely with mammalian M/L cones, and blue and green SCs are most closely related to mammalian S cones. Violet SCs and red DCs are distant relatives of both M/L and S clades, which does not correspond to the sequence relationships of their opsins (*Shichida and Yamashita, 2003*; *Terakita, 2005*) but does correspond to lineages deduced from evolutionary considerations (*Baden and Osorio, 2019*).

HCs (*Figure 11B*): The most notable distinction among HC cells is that some bear axons and other do not. Primate H1, chick HC1 and all mouse HCs are axon-bearing, while primate H2, chick HC2, 4, and 5 do not. However, whereas mammalian axon-bearing and axon-less HCs form separate groups, all chick HCs are outliers.

BCs (*Figure 11C*): The principal division among BCs is into ON and OFF subclasses, based on whether they depolarize or hyperpolarize to light. They form separate clades in each of the three mammalian species taken individually (*Shekhar et al., 2016*; *Peng et al., 2019*; *Yan et al., 2020a*). Chick BC types have not been characterized physiologically, but based on position and key markers that have been validated, we tentatively assigned all 21 types to ON or OFF subclasses, with the 22nd being potentially ON-OFF. Chick putative ON and OFF BCs also generally divide into transcriptomically separate groups (*Figure 5B,C*), but the distinction is less absolute than that for mammals. When all four species are considered together, the distinction remains, but one set of chick ON BCs clusters with most mammalian BC types 4 and 5, leading to a division of ON BCs into two groups.

ACs (*Figure 11—figure supplement 1*): Most if not all ACs use GABA or glycine as a transmitter or co-transmitter. These subclasses form separate clusters when all retinal cells are analyzed together (*Figure 10*) and this distinction is largely maintained when ACs are reanalyzed separately. One type, SACs, form a distinct clade, which includes all five types from all four species (two in chicks) (*Figure 10E* and *Figure 11—figure supplement 1C*). SACs are unique among ACs in many respects: They are the only cholinergic neuron in the retina, they have been found in all vertebrate retinas studied to date, they arise earlier in development than most other ACs, and once generated, their dendrites serve as scaffolds to patterns those of other retinal cell types (*Peng et al., 2017*; *Ray et al., 2018*; *Duan et al., 2018*; *Yan et al., 2020a*). They are transcriptomic outliers among ACs in each species separately; what is noteworthy is that SACs of all species are transcriptomically similar to each other.

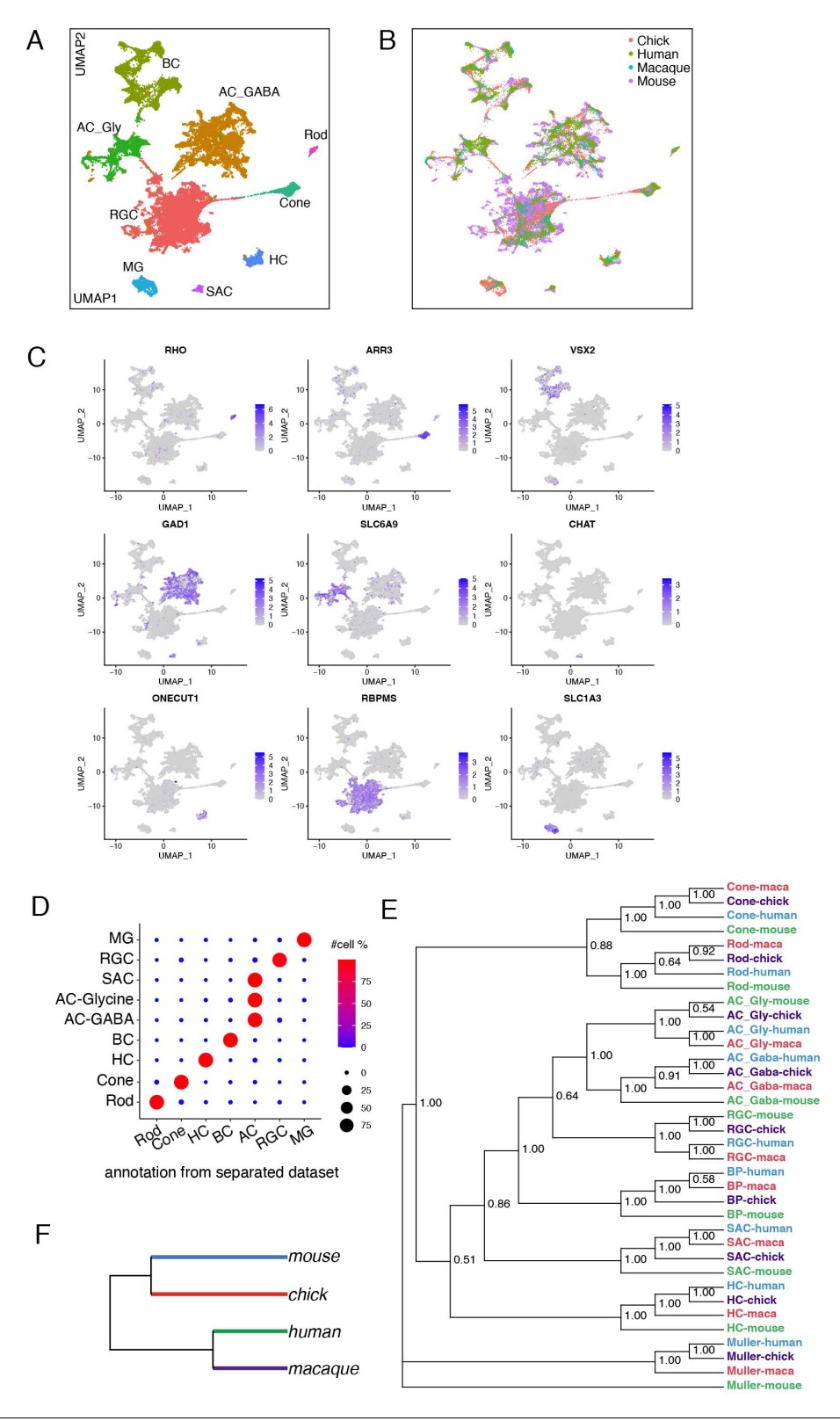

**Figure 10.** Conserved transcriptomic identity of cell classes in mammals and chicks. (**A**) UMAP visualization of pooled cells from chick, mouse, macaque following unsupervised clustering. Colors distinguish classes identified by reference to canonical markers (**C**) and labels previously assigned to each species separately (**D**). (**B**) The same as A, but colors indicate species. (**C**) Feature plot showing the canonical markers of each retina cell class. (**D**) Confusion matrix comparing the class identity by clustering of pooled cells (on y-axis) to the class annotation when each dataset was analyzed

*Figure 10 continued on next page*

*Figure 10 continued*

individually (x-axis). (E) Dendrogram based on the transcriptomic similarity of cell classes from each species. (F) Dendrogram based on overall retinal cell transcriptomic similarity among the four species.

It is difficult to discern similar chick-mammalian conservation for other AC types, at least in part because so few chick AC types have been characterized by structural or physiological criteria. One exception is the VG3 AC, which is identifiable by expression of the glutamate transporter. However, relating clusters derived from a pool of all ACs from all species to types classified in each species separately (*Figure 11—figure supplement 1C*) provides several candidate chick homologs of characterized mammalian AC types.

RGCs (*Figure 11—figure supplement 2*): We showed previously that conservation of types between mouse and primates was lower for RGCs than other cell classes (*Peng et al., 2019*). This relationship extends to chicks. Comparison of the two clustering methods described above shows a conservation of intrinsically photosensitive RGCs, but we detect no chick close relatives of well-studied mammalian RGC types such as midget or parasol RGCs (in primates), direction-selective or alpha RGCs (in mice).

## Discussion

We used scRNA-seq to profile ~40,000 cells from the retina of late embryonic chick retina, at which time all retinal neurons have been born and, as assessed structurally, complex neural circuits have formed (*Prada et al., 1991*; *Cepko et al., 1996*; *Mey and Thanos, 2000*; *Yamagata and Sanes,*

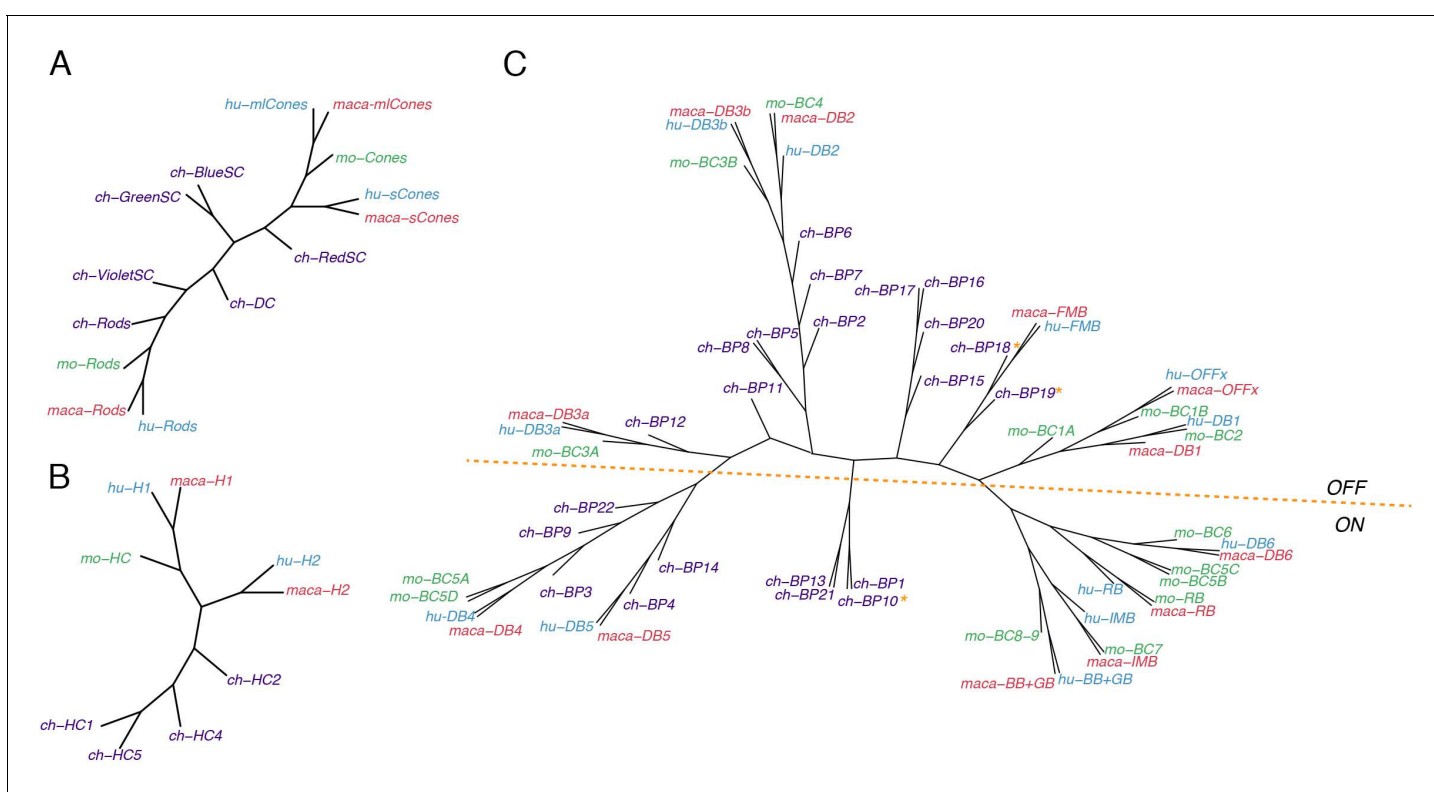

**Figure 11.** Conserved transcriptomic identity of photoreceptor (PR), horizontal (HC), and bipolar cell (BC) types in mammals and chicks. Consensus dendrogram tree for PR (A), HC (B), and BC (C) types from chick, mouse, macaque, and human single-cell dataset. In C, ON and OFF types are globally separated by a dotted line, but three exceptions from chick BC types are indicated by asterisk.

The online version of this article includes the following figure supplement(s) for figure 11:

**Figure supplement 1.** Cross-species comparisons of amacrine cell types.
**Figure supplement 2.** Cross-species comparisons of retinal ganglion cell types.

**Table 1.** Numbers of cell types in mouse, primate (Macaque), and chick retina.

| Cell class | Mouse | Macaque* | Chick (mature) | Chick (developing or topographic) |
|---|---|---|---|---|
| Photoreceptors | 3 | 4 | 8 | 4 |
| Horizontal cells | 1 | 2 | 4 | 1 |
| Bipolar cells | 15 | 12 | 22 | - |
| Amacrine cells | 63 | 34 | 59 | - |
| RGCs | 46 | 18 | 41 | - |
| Müller glia | 1 | 1 | 1 | 6[†] |
| Oligodendrocytes | 0 | 0 | 1 | 4 |
| Astrocytes | 1 | 1 | 0 | 0 |
| Total | 129 | 72 | 136 | 15[†] |

Mouse: **Macosko et al., 2015**; **Shekhar et al., 2016**; **Tran et al., 2019**; **Yan et al., 2020a**.

Macaque: **Peng et al., 2019**.

*Peripheral retinal types plus foveal types not found in periphery.

[†]We found six positional variants of a single MG type, so the number of MG groups is 6, not 5 and the total number of groups is 150 (136+14) not 151 (136 +15).

*1995a*; *Yamagata and Sanes, 1995b*; *Drenhaus et al., 2003*; *Figure 1C*). We used computational methods to divide the cells into 148 clusters or putative cell types (*Table 1*). We then determined the morphology and positions of many types, and in some cases compared them to transcriptomes from cells collected at E12, at which time nearly all retinal neurons have been born but many are immature. We found that 136 of the clusters represent mature retinal types, or potentially small groups of types. The remaining 13 represent topographic variants or developmental intermediates. The mature types comprise what is to our knowledge the first retinal cell atlas for a non-mammalian vertebrate; the topographic variants demonstrate surprisingly long-lasting retention of positional information in Müller glia; and the developmental intermediates reveal stages in postmitotic maturation of several cell types. Finally, we compared the chick atlas to those we recently assembled for mice, monkeys, and humans (*Macosko et al., 2015*; *Shekhar et al., 2016*; *Peng et al., 2019*; *Tran et al., 2019*; *Yan et al., 2020a*; *Yan et al., 2020b*), providing novel insights into the evolution and conservation of neuronal cell classes and types.

## eCHIKIN

Germ-line transgenesis is well established in most commonly used model organisms (mice, zebrafish, *Drosophila*, *C. elegans*) making it possible to generate reporter lines that can be used to reveal the morphology of molecularly defined cell types. This approach has been a powerful one in our studies of mouse and fish retina (*Shekhar et al., 2016*; *Tran et al., 2019*; *Yan et al., 2020a*; *Kölsch et al., 2020*). Lacking this tool for chick, we modified methods for CRISPR-based genome modification in somatic cells (*Mikuni et al., 2016*; *Matsuda and Oinuma, 2019*; *Mikuni, 2020*) to insert reporters or Cre recombinase into genes identified as 'cell-type specific' in our dataset. The use of short (70 bp) homology arms to direct the reporter to appropriate loci made it straightforward to generate targeting fragments. The method can be used to assess the subcellular distribution of the gene product, by fusing the reporter to the coding sequence, or to assess cell morphology, by inserting a soluble fluorescent protein at the translational start site or coupling Cre with a cre-dependent reporter. Of 20 fragements tested, 16 (80%) labeled cells in predicted patterns, including all cases in which we validated the expression pattern by in situ hybridization or antibodies. One potential drawback of the current method is that gene disruption, particularly if both alleles were disrupted, could alter properties of the targeted cells, including their morphology, but we saw no evidence for this in the cells we examined.

## Retinal atlas

The 136 'mature' cell types in our atlas are distributed among seven classes: 8 PRs, 4 HCs, 22 BCs, 59 ACs, 41 RGCs, 1 MG (lumping positional variants, as discussed below), and 1 oligodendrocyte.

The number is substantially greater than that deduced from morphological surveys (11 BCs, 26 RGCs; *Quesada et al., 1988*; *Naito and Chen, 2004*; reviewed in *Seifert et al., 2020*). Nonetheless it may not be complete. First, we did not detect microglia, astrocytes or a variant glial type named diacytes (*Rompani and Cepko, 2010*), even though all are known to be present. Second, we do not know whether our RGCs include displaced types with somata in the INL, although these are known to be present in birds (*Britto et al., 1988*). Third, although the retina is quite mature at E16-18, we cannot exclude the possibility that additional types arise later.

Fourth, the least abundant AC and RGC types we identified comprised 0.4% (28/6642 ACs) and 0.6% (52/8107 RGCs) of the cells in each class. Our sample may have been insufficient to detect still less abundant types: they might have been lumped with related types or missed altogether. As one way to address this possibility, we performed a downsampling test on the three most heterogeneous cell classes – BCs, ACs and RGCs. In this test, one asks how many clusters are obtained from a randomly chosen subset of all cells. Cluster number was unchanged with only 60% of BCs or 80% of RGCs, suggesting that we had profiled enough cells to capture nearly all types in these classes (*Figure 12*). In contrast, the number of AC clusters declined with removal of even 10% of cells, indicating that our dataset was likely too small to capture all types within this class.

*Table 1* compares the number of cell types in the chick atlas to those in mouse, monkey and human. The number of types in chick is similar to those in mouse, and nearly two-fold higher than those in monkey or human. Given the sampling limitation discussed above, however, and the larger number of mouse cells profiled (~140,000 vs ~40,000 in chick) we suspect that the true number of types is higher in chick than in any mammals sampled to date. Thus, our results confirm the suspicion that the chick retina, and perhaps those of other birds, is more complex by this measure than those of mammals.

As noted above, an additional 14 clusters represented developmental or positional variants. We do not include them in the atlas, consistent with current views on the distinction between cell type and cell state (*Zeng and Sanes, 2017*; *Yuste et al., 2020*).

## Retinal cell types

Many of the cell types we identified, such as PRs and HCs, could be matched to types previously characterized morphologically or immunohistochemically (e.g. *Yamagata et al., 2002*; *Yamagata et al., 2006*; *Yamagata and Sanes, 2012*; *Fischer et al., 2007*; *Edqvist et al., 2008*; *Enright et al., 2015*). For nearly all of these, we provide new markers that can be used to learn more about their structure, function, and development of each cell type. For example, HCs types are distinguished by selective expression of receptor-type tyrosine kinases that have been implicated in neuronal differentiation (*LTK, EGFR* and *NTRK1*), and could play roles in their type-specific development or function (*Lemmon and Schlessinger, 2010*). Other intriguing cell types include a putative ON-OFF bipolar type (BC10), a putative serotonergic bipolar type (BC15, expresses the serotonin transporter [*SLC6A2*]; see *Millar et al., 1988*), a glutamatergic amacrine type (AC37, expresses

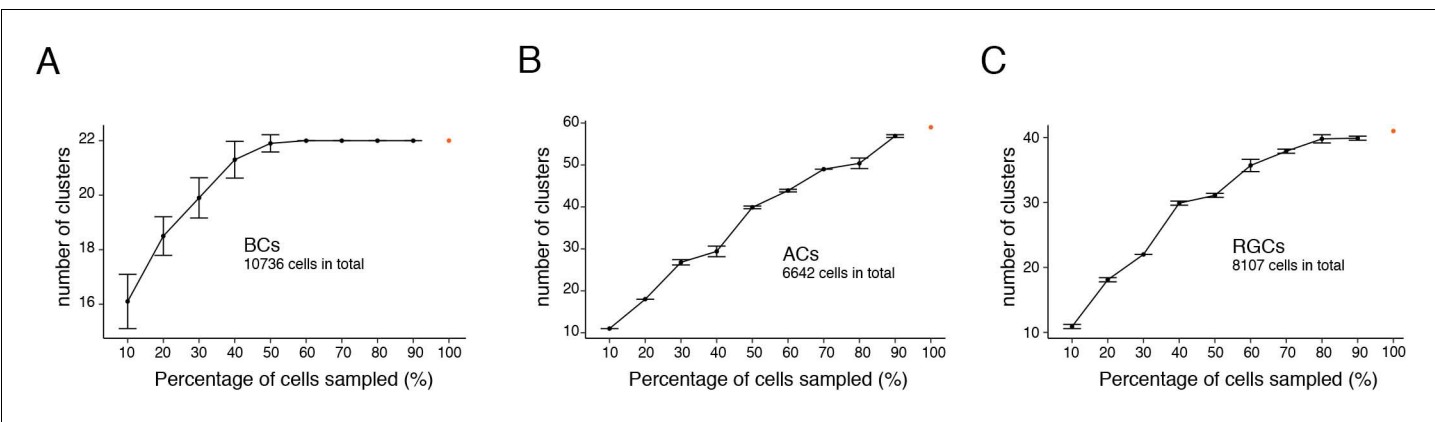

**Figure 12.** Down-sampling test of chick bipolar cells (BCs), amacrine cells (ACs), and RGCs. Graphs show the number of cell clusters identified when using 10–90% of total cells, 10 repeats each (Mean ± SD). (**A**) BCs. (**B**) ACs. (**C**) RGCs.

*VGlut3* [*SLC17A8*]) related to the mammalian VG3 amacrine type (*Krishnaswamy et al., 2015*), and RGC types corresponding to types we showed previously to project to distinct retinorecipient sublaminae in the optic tectum (GC25, B-RGCs; GC1, F-RGCs) (*Yamagata and Sanes, 1995a*; *Yamagata and Sanes, 1995b*). There is a long-standing debate as to whether or not birds may have a 'midget-like' pathway akin to that in the primate retina (*Seifert et al., 2020*). This is an important issue, since midget RGCs comprise 80–90% of all RGCs in humans and other primates. We found a close relationship between chick BC18 and primate 'flat midget bipolars,' one of two BC types that innervate midget RGCs. We did not, however find convincing relationships of any chick RGCs to primate midget RGCs.

## Retained positional information in Müller glia

To date, only a single Müller glial type has been identified in mice and primates using scRNAseq (*Macosko et al., 2015*; *Shekhar et al., 2016*; *Peng et al., 2019*; *Yan et al., 2020b*; *Wang et al., 2017*). In contrast, unsupervised analysis of chick Müller glia generated five clusters, which were, however, less well separated from each other than types in other classes. In situ hybridization with probes for genes differentially expressed among these clusters revealed a positional basis for the heterogeneity. The five clusters were enriched in cells derived from dorsal, ventral, temporal, central, and peripheral retina; and a sixth group, which did not cluster separately, contained cells from nasal retina. Several genes have been shown to exhibit position-dependent expression in early embryos (prior to E8; *Cheng et al., 1995*; *Yuasa et al., 1996*; *Yamagata et al., 1999*; *Sakuta et al., 2001*); in each case, their expression at E18 was consistent with prior data. Five of these six groups appear to be authentically position-dependent, based on studies at earlier ages, while the sixth (peripheral) may largely reflect the central-to-peripheral gradient of retinal development that has been observed in several species (*Kahn, 1974*; *Spence and Robson, 1989*; *Prada et al., 1991*; *Bruhn and Cepko, 1996*).

We sought, but failed to find, positional variants of other cell types. Müller glia are the most abundant single type in our dataset. Thus, weak positional identities may not have been detected in other types. It may be relevant, however, that most of the positional genes we detected were selectively expressed in Müller glia.

Recently, *Hoang et al., 2020* reported single-cell transcriptomic data on P10 chicken retina. Reanalysis of their data suggests that similar graded expression of some of these topographic gene, such as *CHRDL1*, persist after hatching (data not shown). This result is tantalizing in light of recent reports on region- and lamina-selective differences in gene expression of mammalian astrocytes (*Batiuk et al., 2020*; *Bayraktar et al., 2020*). The positional map we describe for chick MG may provide a useful model for investigating the sources and roles of heterogeneity in glial types that were believed until recently to be indivisible.

## Retinal development

Although our aim was to profile mature retinal types, we found 10 clusters that fit within cell classes by our criteria, but lacked mature features: four PRrs, one HC, one Müller glia, and four oligodendrocytes. Several lines of evidence suggested that these clusters were composed of developmental intermediates. First, histological analysis showed that they were more abundant at earlier stages (E10-E12) than at E18 and, in some cases, less abundant still at E20. Second, at intermediate stages, their appearance and disappearance followed the known center to periphery gradient of retinal development (*Kahn, 1974*; *Spence and Robson, 1989*; *Prada et al., 1991*; *Bruhn and Cepko, 1996*) – that is, mature types were more abundant in central than peripheral retina and the opposite was true for the developmental intermediates. Third, we observed a larger proportion of immature PRs and HCs at E12. Finally, in some cases, their gene expression patterns were characteristic of immature cells. For example, immature PRs did not express opsin, but did express genes implicated in PR development (e.g. *ARHGAP18*, *Maeda et al., 2011*; *SLIT1*, *Plump et al., 2002*; *PRDM1*, *Katoh et al., 2010*; *Brzezinski et al., 2013*). Similarly, developing oligodendrocytes expressed genes previously associates with successive stages in oligodendrocyte maturation (*Goldman and Kuypers, 2015*) as well a variety of other genes that now become candidates for stage-specific markers.

A recent study reported on ~5000 single-cell transcriptomes from chick retinas explanted at E1.5–3 and maintained in vitro for 2 days (*Ghinia Tegla et al., 2020*). As expected from the early stage, most of their cells appear to be progenitors, which we did not find at later stages. They did find two cone, two HC, and one RGC cluster, however, consistent with the early birth of these cells, as shown in *Figure 1B*.

### Diversity and evolution of retinal cell types

It has been known since the time of Cajal that the fundamental retinal plan is conserved throughout vertebrates: the same cell classes (PR, HC, BC, AC, RGC, and MG) are present in every species studied to date (*Cajal, 1892*; *Lamb et al., 2007*). Cajal also recognized that in the vast majority of cases, each class is divided into multiple types that differ in morphological detail. Our retinal atlas has now allowed us to address two unanswered questions: Does the resemblance of cell classes across orders extend from morphological to molecular similarity and are types within classes conserved across orders?

The answer to first question is clearly yes. Despite the >300 million years since the mammalian and avian lineages diverged (*Kumar and Hedges, 1998*), all classes remained transcriptomically similar among species, and share expression of key genes that have been used to mark each class in mammals (*Figure 10*). In contrast, types within classes are less well conserved: in only a minority cases, such as PRs, SACs, VG3-ACs and ipRGCs, do chick types have orthologous mammalian types as their closest relatives. This differences between classes and types supports the idea that classes form a common plan within which species evolve distinct types to enable visual behaviors appropriate for their environment, behavioral repertoire, and other sensory capabilities. An open question is whether an underlying similarity might be discovered by examining additional species and/or probing type-specific transcriptional programs. Studies to test these possibilities are underway.

### Conclusions

The chick has been used for thousands of studies on the development, structure and function of the retina and its projections to central targets (*Nicol, 2015*; *Cepko et al., 1996*; *Adler, 2000*; *Mey and Thanos, 2000*; *Wilken and Reh, 2016*; *Wisely et al., 2017*; pubmed search for chick+retina retrieves >4000 papers). Two bottlenecks in moving this work forward have been (a) lack of a global classification and characterization of chick retinal cell types and (b) metrics that can be used to relate chick to mammalian retinal cell types. Our goal in the study reported here has been to address these two challenges. Using the powerful method of high-throughput scRNAseq, we provide the first chick retinal atlas since *Cajal, 1892*, and a gene-specific Golgi-like method to study neuronal morphology, discovering new features of retinal diversity and development as well as insights into the extent to which retinal cell classes and types are conserved between chick and mammals. We hope that our results will facilitate further use of chick as a model for retinal structure, function, and development.

## Materials and methods

**Key resources table**

| Reagent type (species) or resource | Designation | Source or reference | Identifiers | Additional information |
|---|---|---|---|---|
| Genetic reagent (*Gallus gallus*) | GRCg6a | International Chicken Genome Consortium | GCF_000002315.5 | |
| Species | Fertilized chicken eggs (specific pathogen free) | Charles River Laboratories | Cat# 10100326 | |
| Sequenced-based reagent | Sequences of probes used for in situ hybridization | IDT | *Supplementary file 1* in this study | |
| Sequenced-based reagent | Sequences of homology arms used to generate eCHIKIN probes | IDT, this study | *Supplementary file 2* in this study | |
| Sequenced-based reagent | Alt-R CRISPR-Cas9 crRNA (specific to each gene) | IDT | *Supplementary file 2* in this study | |

*Continued on next page*

*Continued*

| Reagent type (species) or resource | Designation | Source or reference | Identifiers | Additional information |
|---|---|---|---|---|
| Recombinant DNA reagent | pXL-BacII-CAG-Zeocin-3xF2A | *Martell et al., 2016* (doi:10.1038/nbt.3563) | | |
| Recombinant DNA reagent | pCAG-PBorf | *Yamagata and Sanes, 2012* (doi:10.1523/JNEUROSCI.3193-12.2012) | | |
| Recombinant DNA reagent | pXL-BacII-CAG-mCherry | This study | | |
| Recombinant DNA reagent | pXL-BacII–loxP-STOP-loxP-Venus | This study | | |
| Recombinant DNA reagent | pXL-BacII-CAG-Venus | *Yamagata and Sanes, 2012* (doi:10.1523/JNEUROSCI.3193-12.2012) | | |
| Recombinant DNA reagent | pCAG-Cre:GFP | Addgene | Addgene#13776 | |
| Antibody | Anti-chicken Thy1 (Mouse monoclonal) | BSJ-1 (*French and Jeffrey, 1986*, doi:10.1002/jnr.490160304) | | 0.01 μ /ml for cell purification |
| Antibody | Anti-Calbindin (Rabbit polyclonal) | Swant | Cat# CB-38a; RRID:AB_10000340 | IF(1/1000) |
| Antibody | Anti-GFP (Rabbit polyclonal) | Millipore | Cat# AB3080P; RRID:AB_2630379 | IF(1/1000) |
| Antibody | Anti-Brn3a (Mouse monoclonal) | Millipore | Cat# MAB1585; RRID:AB_94166 | IF(1/1000) |
| Antibody | Anti-Calretinin (Rabbit polyclonal) | Millipore | Cat# AB5054; RRID:AB_2068506 | IF(1/1000) |
| Antibody | Anti-Calbindin (Rabbit polyclonal) | Swant | Cat# CB38; RRID:AB_10000340 | IF(1/1000) |
| Antibody | Anti-Protein kinase C- α (Rabbit polyclonal) | Sigma | Cat# P4334 | IF(1/10000) |
| Antibody | Anti-VSX2 (Rabbit polyclonal) | GeneTex | Cat# GTX114143 | IF(1/1000) |
| Antibody | Anti-HA tag (Rat monoclonal) | Roche | Clone name: 3F10 | IF(1/1000) |
| Antibody | Anti-Satb1 (Rabbit polyclonal) | Abcam | Cat# ab109122; RRID:AB_10862207 | IF(1/1000) |
| Antibody | Anti-Satb2 (Mouse monoclonal) | Abcam | Cat# ab51502; RRID:AB_882455 | IF(1/1000) |
| Antibody | Anti-Neuropeptide Y (Rabbit polyclonal) | Abcam | Cat# ab10980 | IF(1/1000) |
| Antibody | Anti-AP2A (Mouse monoclonal) | Developmental Studies Hybridoma Bank | Clone name: 3B5 | IF(1/100) |
| Antibody | Anti-AP2B (Mouse monoclonal) | Developmental Studies Hybridoma Bank | Clone name: 2A4 | IF(1/100) |
| Antibody | Anti-OTX1 (Mouse monoclonal) | Developmental Studies Hybridoma Bank | Clone name: 5F5 | IF(1/100) |

*Continued on next page*

*Continued*

| Reagent type (species) or resource | Designation | Source or reference | Identifiers | Additional information |
|---|---|---|---|---|
| Antibody | Anti-SOX5 (Mouse monoclonal) | Developmental Studies Hybridoma Bank | Clone name: 1C12 | IF(1/100) |
| Antibody | Anti-PAX6 (Mouse monoclonal) | Developmental Studies Hybridoma Bank | Clone name: PAX6 | IF(1/100) |
| Antibody | Anti-NMB (Mouse monoclonal) | Developmental Studies Hybridoma Bank | Clone name: NMB1 | IF(1/10) |
| Antibody | Anti-STRA6 (Mouse polyclonal) | This study | | IF (1:1000) |
| Antibody | Anti-TPBGL (Mouse polyclonal) | This study | | IF (1:500) |
| Antibody | Anti-SLC6A4 (Mouse polyclonal) | This study | | IF (1:200) |
| Antibody | Anti-chicken choline acetyltransferase (Rabbit polyclonal) | *Johnson and Epstein, 1986* (doi:10.1111/j.1471-4159.1986.tb13064.x) | | IF (1:1000) |
| Antibody | Anti-chicken glutamate synthetase (Rabbit polyclonal) | *Linser and Moscona, 1979* (doi: 10.1073/pnas.76.12.6476) | | IF (1:1000) |
| Commercial assay or kit | Papain Dissociation System, Without EBSS | Worthington | Cat# LK003160 | |
| Commercial assay or kit | Chromium Single Cell 30Library and Gel Bead Kit v2, 10X Genomics Cat#120237 16rxns | 10 X Genomics | Cat# 120237 | |
| Chemical compound, drug | Goat anti-mouse IgG conjugated magnetic beads | Miltenyi Biotec | Cat# 484–02 | |
| Chemical compound, drug | Alt-R CRISPR-Cas9 tracrRNA | IDT | Cat# 1072533 | |
| Chemical compound, drug | Alt-R S.p. Cas9 Nuclease V3 | IDT | Cat# 1081058 | |
| Chemical compound, drug | Alt-R Cas9 Electroporation Enhancer | IDT | Cat# 1075915 | |
| Chemical compound, drug | Alt-R HDR Enhancer | IDT | Cat# 1081072 | |
| Chemical compound, drug | Fast Green FCF | Sigma-Aldrich | Cat# F7252 | |
| Commercial assay or kit | TSA Cyanine 3 Plus Evaluation Kit | Perkin Elmer | Cat# NEL744E001KT (FP1170) | |
| Commercial assay or kit | TSA Fluorescein Plus Evaluation Kit | Perkin Elmer | Cat# NEL741E001KT (FP1168) | |
| Chemical compound, drug | Anti-Digoxigenin-POD, Fab fragments | Roche | Cat# 11207733910 | |
| Commercial assay or kit | EconoTaq PLUS GREEN 2X Master Mix | Lucigen | Cat# 30033–1 | |
| Commercial assay or kit | CIAquick-Gel Extraction Kit | Oiagen | Cat# 28704 | |
| Commercial assay or kit | MiniMACS Separation Unit | Miltenyi Biotec | Cat# 421–02 | |
| Other | MACS Multistand | Miltenyi Biotec | Cat# 423–03 | |
| Other | Type WHM large cell separation columns | Miltenyi Biotec | Cat# 422–02 | |

*Continued on next page*

*Continued*

| Reagent type (species) or resource | Designation | Source or reference | Identifiers | Additional information |
|---|---|---|---|---|
| Other | BTX 830 | BTX | Cat# 45–0662 | |
| Other | Genetrode | BTX | Cat# 45–0116 | |
| Software, algorithm | ImageJ (Fiji) Version 2.1.0 | Fiji | https://imagej.net/Fiji | |
| Software, algorithm | R 3.6.2 | The R foundation | https://www.r-project.org/ | |
| Software, algorithm | RStudio 1.3.1056 | Rstudio | https://rstudio.com | |
| Software, algorithm | Adobe Photoshop 20.0.9 release | Adobe | https://www.adobe.com | |

## Single-cell RNA-seq

Animals were used in accordance with NIH guidelines and protocols approved by Institutional Animal Use and Care Committee at Harvard University. Fertilized chicken eggs (specific pathogen free) were obtained from Charles River Laboratories (Wilmington, MA), and incubated in a 1550 HATCHER (GQF MFG, Savannah, GA) at 38°C. Retinas were dissected in Hanks' balanced salt solution supplemented with 20 mM HEPES, pH 7.4 (HBSS). After removing pigment epithelial cells and pecten, whole retinae from both eyes were dissociated at 37°C for 30 min with papain (LK003160, Worthington, Lakewood, NJ). The same volume of Neurobasal medium (Thermo Fischer, Waltham, MA) and 10 µg of deoxyribonuclease I (DN25, Sigma, St. Louis, MO) were added, and cells were triturated. To remove debris, cells were then washed twice through a cushion of the ovomucoid protease inhibitor/BSA/HBSS in the papain dissociation medium. Cells were counted, resuspended in PBS with acetylated BSA, and processed with the Chromium Next GEM Single Cell 3' Library Construction Kit (version 2) (10x Genomics, Pleasanton, CA; *Zheng et al., 2017*). Briefly, single cells are partitioned into oil droplets containing single oligonucleotide-derivatized beads followed by cell lysis, barcoded reverse transcription of RNA, amplification, shearing, and attachment of 5' adaptor and sample index oligos. Libraries were sequenced on the Illumina HiSeq 2500 (Paired end reads: Read 1, 26 bp, Read 2, 98 bp).

To enrich RGCs, cells dissociated as above were immunopurified with mouse monoclonal antibody to Thy-1 (BSJ-1: *French and Jeffrey, 1986*) using goat anti-mouse IgG conjugated magnetic beads (Miltenyi Biotec, Auburn, CA) (*Yamagata et al., 2002*). Cells were then resuspended in PBS with acetylated BSA and processed as above.

## Analysis of scRNA-seq data

We analyzed scRNA-seq data using a pipeline modified from *Peng et al., 2019*. Steps are as follows:

1. Sample demultiplexing was performed with cellranger mkfastq function (10X Genomics, version 2.1.0), and reads were aligned to the reference genome GRCg6a using cellranger count function (10X Genomics, version 2.1.0) with the option `-force-cells`=8000. A threshold of 600 genes detected per cell was applied to filter out low quality cells and debris.
2. Clustering was performed using the R package 'Seurat' to stratify cells into major classes using defining class-selective markers (*Stuart et al., 2019*; *McInnes and Healy, 2018*). Every class was then reanalyzed individually to maximize the dynamic range in feature space for each cell class.
   (2.1) Unique molecular identifiers (UMIs) count were log-normalized to calculate the expression level using the 'LogNormalize' method with a scale factor of 10000 in the 'Seurat' function 'NormalizeData'.
   (2.2) Highly variable genes were selected by the method from *Pandey et al., 2018* and used for further clustering.
   (2.3) Principal component (PC) analysis was performed using the Seurat function RunPCA. Significant PCs were estimated based on the Tracy-Widom theory (*Patterson et al., 2006*) and used for further analysis.
   (2.4) K-nearest neighbor graph was constructed using 'Seurat' function FindNeighbors with the number of significant PCs identified from 2.3.

(2.5) Clusters were identified using 'Seurat' function FindClusters with resolution of 0.8. We used a relatively fine resolution to generate the initial clusters, then refined them based on the number of differentially expressed (DE) genes as described below to prevent over-clustering.
(2.6) Uniform Manifold Approximation and Projection (UMAP) visualization was generated using 'Seurat' function RunUMAP with the number of significant PCs identified from 2.3.

3. Because some contaminants became evident only following division into classes, we reexamined the clustered data to remove contaminant-like clusters and doublets. The quality of each cluster was assessed by transcriptomic complexity and expression of typical house-keeping genes (*Eisenberg and Levanon, 2013*), and low-quality cells were removed. Doublets were identified as cells that co-expresses genes from multiple classes, lacked uniquely expressed marker genes, and/or exhibited >1.5 fold higher number of UMIs than other clusters in the same cell class.

4. Clusters closely related on the dendrogram were assessed using a generalized linear model from the R package 'MAST' (*Finak et al., 2015*), and iteratively merged if no DE genes were found.

Trajectory analysis of oligodendrocytes was performed using the R package 'Monocle3' (*Cao et al., 2019*).

For cross species analysis, types from each species were down-sampled to a maximal of 200 before pooling. Genes of non-human species were converted to their human homologene using the R package 'homologene' and only those with homologenes in all four species were retained for analysis. Clustering was performed using 'Seurat' as described above, but with cells from different species aligned using the 'Reference-based' integration with all four species as reference. Dendrograms were then built using scaled data after integration, with a total of 100 trees in each case; 80% of cells and 80% of the highly variable genes were used in each build. Final trees were generated using the 'consensus' function from the 'Clann' package (*Creevey and McInerney, 2005*; *Figure 10F*, *Figure 11* and y-axis of *Figure 11—figure supplements 1C* and *2C*). Alternatively, trees were built from clusters identified in the integrated space (x-axis of *Figure 11—figure supplements 1C* and *2C*).

Initial assessment of the aligned data showed a complete absence of violet opsin (*OPN1SW*), which we found to result from incomplete sequence in the genome file GRCg6a. We therefore added the cDNA sequence to the annotation file.

A down-sampling test was performed by randomly sampling 10–90% of cells within each class in steps of 10% of the total number of cells used to generate the atlas. At each step, downsampling was performed 10 times, using different randomly selected subsets. Subsets were then clustered by the same methods as described above.

## Antibodies and immunostaining

Antibodies used in this study were: rabbit polyclonal antibodies to GFP (Millipore, AB3080P); mouse monoclonal anti-Brn3a (clone, 5A3.2, Millipore, MAB1585); rabbit anti-calretinin (Millipore, AB5054); rabbit anti-calbindin (Swant, CB38); rabbit anti-protein kinase C- α (PKCα) (Sigma, P4334); rabbit anti-VSX2 (Chx10, GeneTex, GTX114143); rat monoclonal anti-HA tag (Roche, 3F10); rabbit anti-Satb1(Abcam, ab109122); mouse monoclonal anti-Satb2 (Abcam, ab51502); rabbit anti-Met-enkephalin (ImmunoStar, 20065); rabbit anti-neuropeptide Y (Abcam ab10980). Mouse monoclonal antibodies AP2A (clone, 3B5), OTX1 (Otx-5F5), and PAX6 were from Developmental Studies Hybridoma Bank (Iowa City, IA). A mouse monoclonal antibody to neuromedin-B (NMB1) was previously described (*Yamagata et al., 2006*) and is available from Developmental Studies Hybridoma Bank.

Antibodies to cell surface proteins were generated by immunizing L cells that had been stably transfected with cDNAs (*Yamagata and Sanes, 2012*). To generate stable cell lines, full-length cDNAs were generated using SuperScript III (Thermo Fisher), amplified from cDNA obtained from chick retina using a high-fidelity Phusion DNA polymerase (NEB), cloned into a piggyBac transposon vector pXL-CAG-Zeocin-3xF2A (NotI/AscI sites; *Martell et al., 2016*) by Gibson assembly (NEBuilder HiFi DNA Assembly, NEB), transfected to L cells together with a piggyBac transposase vector pCAG-PBorf using Transporter five transfection reagent (Polyscience), and selected in 1 mg/ml Zeocin (Invivogen) for 10–14 days. Cells resistant to Zeocin were pooled, grown, harvested, and rinsed with PBS three times. Female mice were then injected intraperitoneally with $10^7$ cells in 0.5 ml four to five times at 2–3 weeks intervals, beginning at 6 weeks of age. Antiserum was then collected,

incubated with paraformaldehyde-fixed untransfected L cells to remove irrelevant antibodies, and used for immunostaining. To validate antibodies, 293T human embryonic kidney cells were transfected with each construct, and immunostained. In each case, the antibody stained cells transfected with a cDNA encoding the cognate immunogen but not untransfected cells or cells transfected with cDNA encoding other immunogents (data not shown).

For immunostaining, retinas were fixed with 4% (w/v) paraformaldehyde/PBS overnight at 4℃, sunk in 15%(w/v) and 30%(w/v) sucrose/PBS, and mounted in Tissue Freezing Medium (EM Sciences, Hatfield, PA). Sections were cut in a cryostat, either permeabilized with 0.1% (w/v) TritonX-100/PBS for 5 min at room temperature or with 100% methanol for 15 min at −20℃, blocked with 5% (w/v) skim milk/PBS for 30 min at room temperature, incubated with appropriate antibodies overnight, rinsed, and incubated with appropriate secondary antibodies (Jackson ImmunoResearch, West Grove, PA) and NeuroTrace 640 (ThermoFisher/Invitrogen). After rinsing with PBS, sections were mounted in VECTASHIELD (Vector Labs, Burlingame, CA) and imaged with a Zeiss Meta510 confocal microscope (Oberkochen, Germany).

In situ hybridization cDNA for generating RNA probes were amplified from cDNA obtained from chick retina using SuperScript III (Thermo Fisher) and a high-fidelity Phusion DNA polymerase (NEB) and cloned into either pCMV vectors using Gibson Assembly or into pCR2.1 TOPO (Thermo Fisher) by TOPO cloning. Probe sequences are listed in *Supplementary file 1*.

RNA probes were generated from the linearized plasmids using T7 RNA polymerase (Thermo Fisher) and digoxygenin- or fluorescein-labeled nucleotides (Roche), and hydrolyzed to around 500 bp if needed. In situ hybridization using nitro-blue tetrazolium and 5-bromo-4-chloro-3′-indolyphosphate and double color in situ hybridization using TSA Plus (PerkinElmer) were performed as previously described (*Yamagata et al., 1999*; *Yamagata and Sanes, 2012*). For double-color in situ hybridization, slides were incubated with 0.1M glycine-HCl, pH 2.0, for 30 min at room temperature following the first color reaction to remove peroxidase-conjugated anti-hapten antibodies and thereby avoid cross-reactivity. Sections were mounted in and imaged with a Zeiss Meta510 confocal microscope.

## eCHIKIN

To label and visualize cells that express marker genes, we devised eCHIKIN (*e*lectroporation- and *C*RISPR-mediated *H*omology-*I*nstructed *K*nock-*IN*). nBriefly, our method resembles in several respects two others that adapted the initial SLENDR technology (*Mikuni et al., 2016*) to mouse embryos (*Ohtsuka et al., 2018*; *Miura et al., 2018*; *Gurumurthy et al., 2019*). We introduced CRISPR/Cas9 ribonucleoprotein complexes and single-strand DNA to chick embryos by in ovo electroporation, using reagents from the Alt-R CRISPR-Cas9 System (IDT, Coralville, IA). Cas9 requires a CRISPR RNA (crRNA) to specify the DNA target sequence. The crRNA sequence was designed to target the sequence near the initiation codon (ATG) based on *S. pyogenes* PAM sequence and the MIT guide specificity score in the UCSC genome browser (https://genome.ucsc.edu) (*Supplementary file 2*). The crRNA (0.1nmole) was first annealed with an equimolar amount of trans-activating crRNA (tracrRNA) in 5 µl in the annealing buffer (GenScript) by heating at 95℃ for 5 min followed by rapid chilling. This product was then incubated with *S. pyogenes* Cas9 protein (5 µg) for 60 min at room temperature to prepare a ribonucleoprotein complex. This complex was mixed with a single-strand DNA (0.1–0.5 µg) and the other components described below. Each single-strand DNA contains ~70 base gene-specific homology arms at the both ends (*Supplementary file 2*). Short single strand DNAs were purchased from IDT (Coralville, Iowa). Longer single strand DNA was prepared by asymmentric PCR using a 1:100 ratio of two primers. Templates included Venus (*Aequorea coerulescens* GFP variant) or Cre (Addgene plasmid #14797). Following amplification with EconoTaq PLUS GREEN 2X Master Mix (Lucigen), products were agarose gel-purified to select single strand DNA using the QiaQuick Gel Extraction kit (Qiagen); 20%(v/v) isopropanol was added to the solubilization solution before binding to spin columns.

The mixture of ribonucleoprotein and single-strand DNA (0.1–0.5 µg) was electroporated to developing chick embryos (Hamburger-Hamilton stage ~10, E1.5) together with 1 µM Cas9 electroporation enhancer (carrier DNA from IDT) and 0.1 mM HDR enhancer (DNA ligase IV inhibitor from IDT) to enhance homologous recombination. We also added 1 µg piggyBac transposon reporter (pXL-CAG-mCherry) and 0.1 µg a transposase construct (pCAG-PBorf) to monitor successful electroporation. For Cre activation, 1 µg of pXL-CAG-loxP-STOP-loxP-Venus was also included. All the

DNA reagents were prepared in advance as ribonuclease-free by extensive phenol-chloroform extractions followed by ethanol precipitation and rinsing with 70%(v/v) ethanol. A total of 0.01%(w/v) Fast Green (Sigma) was added to monitor injection.

Electroporation was with six square pulses of 7 V for 25 ms using ECM830 (Harvard Apparatus) after immersing electrodes with Hanks' balanced salt solution supplemented with 50 µg/ml kanamycin. After sealing eggshells with plastic tapes, eggs were returned to 37°C incubator.

## Image and statistical analysis

Images were processed with Adobe Photoshop, and Image-J (Version 1.47d, Fiji). Position of spots were measured using Image-J. Single-cell RNA-Seq data were analyzed using R 3.6.2 (The R foundation, https://www.r-project.org/).

## Acknowledgements

This work was supported by grants R01EY022073 and R37NS029169 from the NIH. We thank Mallory A Laboulaye for participation in initial studies and Karthik Shekhar for advice.

## Additional information

### Funding

| Funder | Grant reference number | Author |
| --- | --- | --- |
| National Eye Institute | RO1EY022073 | Masahito Yamagata<br>Wenjun Yan<br>Joshua R Sanes |
| National Institute of Neurological Disorders and Stroke | R37NS029269 | Masahito Yamagata<br>Wenjun Yan<br>Joshua R Sanes |

The funders had no role in study design, data collection and interpretation, or the decision to submit the work for publication.

### Author contributions

Masahito Yamagata, Conceptualization, Validation, Investigation, Visualization, Methodology, Writing - original draft, Writing - review and editing; Wenjun Yan, Conceptualization, Data curation, Software, Formal analysis, Visualization, Methodology, Writing - original draft; Joshua R Sanes, Conceptualization, Funding acquisition, Writing - original draft, Project administration, Writing - review and editing

### Author ORCIDs

Masahito Yamagata (iD) https://orcid.org/0000-0001-8193-2931
Wenjun Yan (iD) https://orcid.org/0000-0003-3568-4265
Joshua R Sanes (iD) https://orcid.org/0000-0001-8926-8836

### Ethics

Animal experimentation: This study was performed in strict accordance with the recommendations in the Guide for the Care and Use of Laboratory Animals of the National Institutes of Health. All of the animals were handled according to approved institutional animal care and use committee (IACUC) protocols (#24-10) of Harvard University.

### Decision letter and Author response

Decision letter https://doi.org/10.7554/eLife.63907.sa1
Author response https://doi.org/10.7554/eLife.63907.sa2

## Additional files

### Supplementary files

- Supplementary file 1. Sequences of probes used for in situ hybridization.
- Supplementary file 2. Sequences of homology arms used to generate eCHIKIN probes.
- Supplementary file 3. Basic quality measures of each cluster.
- Transparent reporting form

### Data availability

Sequencing data have been deposited in GEO under accession GSE159107. Data can be visualized at the Broad Institute Single Cell Portal using the link: https://singlecell.broadinstitute.org/single_cell/study/SCP1159.

The following datasets were generated:

| Author(s) | Year | Dataset title | Dataset URL | Database and Identifier |
|---|---|---|---|---|
| Yamagata M, Yan W, Sanes JR | 2021 | Cell Atlas Of The Chick Retina: Single Cell Profiling Identifies 150 Cell Types | https://www.ncbi.nlm.nih.gov/geo/query/acc.cgi?acc=GSE159107 | NCBI Gene Expression Omnibus, GSE159107 |
| Yamagata M, Yan W, Sanes JR | 2021 | A cell atlas of the chick retina based on single-cell transcriptomics | https://singlecell.broad-institute.org/single_cell/study/SCP1159 | Broad Single Cell Portal, SCP1159 |

The following previously published datasets were used:

| Author(s) | Year | Dataset title | Dataset URL | Database and Identifier |
|---|---|---|---|---|
| Macosko EZ | 2015 | Drop-Seq analysis of P14 mouse retina single-cell suspension | https://www.ncbi.nlm.nih.gov/geo/query/acc.cgi?acc=GSE63472 | NCBI Gene Expression Omnibus, GSE63472 |
| Shekhar K | 2016 | Drop-Seq analysis of P17 FACS sorted retinal cells from the Tg (Chx10-EGFP/cre,-ALPP)2Clc or Vsx2-GFP transgenic line | https://www.ncbi.nlm.nih.gov/geo/query/acc.cgi?acc=GSE81904 | NCBI Gene Expression Omnibus, GSE81904 |
| Tan NM | 2019 | Single-cell profiles of retinal neurons differing in resilience to injury reveal neuroprotective genes | https://www.ncbi.nlm.nih.gov/geo/query/acc.cgi?acc=GSE137400 | NCBI Gene Expression Omnibus, GSE137400 |
| Peng Y, Shekhar K, Yan W, Do MT, Regev A, Sanes JR | 2019 | Molecular specification of cell types underlying central and peripheral vision in primates (macaque peripheral single cell RNA-seq) | https://www.ncbi.nlm.nih.gov/geo/query/acc.cgi?acc=GSE118852 | NCBI Gene Expression Omnibus, GSE118852 |
| Yan W, Laboulaye MA, Tran NM, Whitney IE, Benhar I, Sanes JR | 2020 | Mouse retinal cell atlas: molecular identification of sixty-three amacrine cell types | https://www.ncbi.nlm.nih.gov/geo/query/acc.cgi?acc=GSE149715 | NCBI Gene Expression Omnibus, GSE149715 |
| Yan W, Peng Y, van Zyl T, Regev A, Shekhar K, Juric D, Sanes JR | 2020 | Cell atlas of the human fovea and peripheral retina | https://www.ncbi.nlm.nih.gov/geo/query/acc.cgi?acc=GSE148077 | NCBI Gene Expression Omnibus, GSE148077 |

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
