## [Decision Letter]

**Acceptance summary:**

Your paper profiling the large diversity of chick retinal neurons taking advantage of single-cell mRNA sequencing is novel and exciting and this dataset published in *eLife* will be an important resource for retina scientists. Furthermore, the comparison with the mouse retina is of great interest and the variation among the retinal ganglion cells might represent a fundamental process by which species adapt to their visual environment.

**Decision letter after peer review:**

Thank you for submitting your article "Single cell profiling identifies 136 cell types in the chick retina" for consideration by *eLife*. Your article has been reviewed by three peer reviewers, and the evaluation has been overseen by a Reviewing Editor and Marianne Bronner as the Senior Editor. The following individual involved in review of your submission has agreed to reveal their identity: Tom Baden (Reviewer #2).

The reviewers have discussed the reviews with one another and the Reviewing Editor has drafted this decision to help you prepare a revised submission.

The three reviewers found your paper novel and exciting and they are anxious to see it published in *eLife* as this will be an important resource for the field. Furthermore, the comparison with the mouse retina is of great interest and the variation among the retinal ganglion cells might represent a fundamental process by which species adapt to their visual environment.

However, there are a few points that you will need to address before the paper can be published, most of which should only require editorial work or bioinformatic processing.

– Present a much better description of the technical processing the scRNA seq data.

– Address the question of cell type: All reviewers feel that you have not demonstrated that you have identified 136 cell types as several could be cell states. One way to address this would be to add a pseudotime analysis to make sure that the “cell types” are not different cell states.

– The cell types that differ between species might be due to the very different stages of development of the various retinas. Showing a UMAP of all ages aggregated together would address this point.

– You should also fix the referencing to other papers that appears to be lacking.

We will be expecting a revised version soon and the field will then be able to enjoy these results.

Reviewer #1:

The manuscript "Single cell profiling identifies 136 cell types in the chick retina" by Yamagata and colleagues represents a new transcriptomic atlas of the chicken retina. It explores the diversity of cell types in the avian retina, describing the functional significance of this vast diversity in terms of cell morphology, connections and spatial organization within the retina. Using a CRISPR-based assay, they show the position of the various cell types and their corresponding morphologies. Finally, the authors describe the regionalization of specific populations of glial cells in their respective identities. Overall this manuscript is an extensive documentation combining in silico exploration with in vivo validations. It is an impressive resource that will be extremely valuable to the field. While the experimental design of the study is correct and in general the data appears technically rigorous, the Results section lacks clarity in the way the data are presented, and more importantly, it does not provide sufficient information on the methods and the parameters used for the single-cell analysis. Altogether their findings appear convincing and provide an impressive resource on which the community will be able to generate new hypotheses in the field.

1) The central point that is made by the authors regards the number of cell types in the avian retina. Based on previous reports in other species, this number may be expected. But how much of the diversity one can observe in amacrine and bipolar cells is related to the transitional states of the same cell type? Also, the authors mention 10 clearly immature clusters: could these correspond to CMZ-derived or CMZ cells? Finally, even if the retina at E18 is histologically mature, some of the cells in mature clusters may be closer to full maturation than others, in particular when it comes to synapse refinement and pruning. Was this taken into account? A first step into this investigation would be to rank the cells with pseudotime alignment method (such as Monocle).

2) To what extent is the number of cell types related to i) the number of single-cells sequenced (have you controlled whether you are close to saturation?) ii) the stages used to evaluate them (E12-E18) and iii) the clustering technique (this latter was difficult to assess due to insufficient description) ?

3) Why is there no dimensionality reduction visualization of the main dataset? We see these cells aggregated with other species only in Figure 10, but nowhere can we appreciate the distinction between the main classes. Thus, it is hard to assess to what extent these classes are distinct.

4) The manuscript, presented as a resource, occasionally lacks clarity in its presentation, especially in the first pages of the Results section (Figures 1 and 2).

5) The main concern is the lack of detail in the Materials and methods section, in particular on the single-cell analysis (QCs, etc). The clustering appears to be correct, but it would be valuable to add how filtering and hierarchical clustering were performed. Also, a clarification of the rationale for the maximisation procedure used to distinguish cell types would be useful. Same for the thresholds used.

6) Regarding the data availability, the link provided does not work:

"https://singlecell.broadinstitute.org/single_cell/study/SCP1159"

7) The Introduction and Discussion are well written but poorly referenced, especially when it comes to the multi-species comparison. As this study adds to an increasing body of work on single-cell retina, it seems important to refer to the previous work (unless the number of references is limited?).

Reviewer #2:

In their manuscript "Single cell profiling identifies 136 cell types in the chick retina", Yamagata, Yan and Sanes present the first transcriptomic atlas of an avian retina. The manuscript delivers pretty much what it says on the tin: a (probably almost) complete parts-list of the chicken retina, with 6 photoreceptor types, 4 HCs, >20 BCs, >40 RGCs and close to 60 ACs – alongside Muller glia. Each of these comes with a hit-list of molecular markers than could be used to genetically target them in the future, and some are in fact confirmed in the study using their novel eCHIKIN approach.

Beyond this, the authors also compare their results from chicken to their previous results from mouse as well as species of primate including humans and find interesting links between cell classes and types. Overall, their results confirm several long-held notions, for example that avian retinas are among the more complex ones out there, that retinal cell homology works incredibly well on the retinal input side (especially HCs seem to be amazingly conserved!), while this works less well on the RGC side – possibly hinting that evolution will more readily tweak the retinal output than the input?

Finally, the work is technically excellent, the manuscript is written clearly and succinctly, and referencing is adequate.

Overall, there is a lot to like, and really nothing to dislike. As a lab that is actively investigating vision in chicks (and other species), I can attest first-hand that this dataset is incredibly useful and interesting, and I enthusiastically support publication in e*Life*.

Reviewer #3:

In this descriptive study the authors use scRNA-Seq to profile ~40,000 cells from the embryonic chick retina and compare them to similar datasets generated from mouse, human, and macaque. As the authors point out, the chick retina is a long-used model for retinal development and recently another paper described scRNA-Seq on the chick retina in the context of multi-species functional studies of retinal regeneration (Hoang et al., 2020). An additional study (Tegla et al., 2020) also performed scRNA-Seq of the chick retina to probe the function of developmental transcriptional programs. The field does not however have a well-documented single cell atlas of the chick retina. This current study represents such a resource that will be an important reference for the field moving forward.

1) Throughout the paper the authors associate the concept of single-cell cluster with the concept of cell type. Given that the number and size of clusters can be dynamically changed depending on analysis parameters, I would strongly recommend that the authors avoid using the term "cell type", except where this is well supported experimentally (e.g. by demonstration of a cell-mosaic arrangement of marker expression). This term is especially misleading in the title, which might more accurately be "a single cell atlas of the developing chick retina".

2) The description of the scRNA-seq analyses is really insufficient to permit replication or in some cases interpretation of the data. The authors need to give more details with regard to the specific parameters used for each analysis.

3) The authors should provide additional evidence that comparing cell classes and types between the embryonic chick retina to that of the more mature mammalian retinas is experimentally justified. Could it be that they are seeing a greater diversity of cell types in the chick retina because they are considering a greater diversity of transitional states? These multi-species comparisons would be greatly strengthened by addition of a dataset from the mature chicken retina.

---

## [Author Response]

[…] There are a few points that you will need to address before the paper can be published, most of which should only require editorial work or bioinformatic processing.– Present a much better description of the technical processing the scRNA seq data.

We have expanded the methods section as requested, and include all the key parameters. We have also added a table with quality control metrics.

– Address the question of cell type: All reviewers feel that you have not demonstrated that you have identified 136 cell types as several could be cell states. One way to address this would be to add a pseudotime analysis to make sure that the “cell types” are not different cell states.

We believe that this issue reflects a misunderstanding of how we generated the atlas. It arises from the regrettable way in which we presented Figure 1. Our aim was to summarize, in a single figure, all the single cell datasets we used. It therefore gives a mistaken impression of which cells were used to generate the atlas and which were either discarded or used for other purposes. We have revised the figure, figure legend and accompanying text for clarity and summarize here the main points:

1) Some of reviewer 1’s comments seem to assume that we combined multiple time points in generating the atlas. This is not correct. We derived our census of types from a single stage: E16 for RGCs and E18 for all other cells. The E16 data were not used to derive types for any classes other than RGCs, the E18 data were not used in generating the RGC atlas, and the E12 data were used only to test developmental hypotheses – most prominently seeking developmental intermediates so we could distinguish them from authentic types. As noted above, we have revised the figure and text to avoid any confusion on this point. We note in our defense, however, that reviewers 2 and 3 did not seem to be confused on these points.

2) Had we used multiple time points to generate the atlas, a pseudotime analysis could be valuable. However, since each type derives from a single time point, this issue is moot. Performing pseudotime analysis on this dataset would not clearly distinguish states of a single type from closely related types.

3) There are indeed some positional and developmental variants included in our dataset, but we have been careful to distinguish them from types. Altogether, we found 150 clusters, of which 136 correspond to types (or potentially a few types lumped together) and the other 14 correspond to states: positional variants for Müller glia and developmental intermediates for photoreceptors, horizontal cells and oligodendrocytes. Table 1 makes the distinction clear and we have added text to the Introduction, Results and Discussion section to be sure there is no residual confusion.

4) The title states that we identified 136 types, a statement we stand by. We did not say that there are only 136 types, just as a paper with the title “Identification of 4 novel genes” does not imply that there are only 4 novel genes to be found. Nonetheless, to avoid controversy, we have altered the title to “A cell atlas of the chick retina based on single cell transcriptomics”

5) Regarding nomenclature, reviewer 2 raises no issues with respect to the type vs. state distinction while reviewer 3 makes the radical suggestion that we abandon the term “cell type” altogether. Here, we respectfully decline. The idea of cell types is firmly established in the literature, and we and others have discussed in detail the thorny issue of distinguishing types from states (e.g., Zeng and Sanes, 2017; Yuste et al., 2020). That is not to say it is a fully solved problem – but the term is now in common use in studies of rodents, primates, fish and flies, and avoiding it leads to nothing but confusion. Instead, we have expanded the Discussion to reconsider the issue and make our criteria clear.

6) A small point: reviewer 2 did not raise the state/type issue.

– The cell types that differ between species might be due to the very different stages of development of the various retinas. Showing a UMAP of all ages aggregated together would address this point.

We believe that this comment, again from reviewer 1, arises from the same misunderstanding considered above. We derived our census of types from mature retinas: E16 for RGCs and E18 for all other cells. Thus, showing a UMAP combining all ages would not be helpful. In response to a related suggestion below we have, however, added a UMAP of the E16+E18 data to Figure 1.

– You should also fix the referencing to other papers that appears to be lacking.

We have added references as requested. We also cite a comprehensive review of transcriptomically based retinal cell atlases, now in press (Shekhar and Sanes, “Generating and using transcriptomically based retinal cell atlases”, Annual Review of Vision Science)

We will be expecting a revised version soon and the field will then be able to enjoy these results!Reviewer #1:[…] 1) The central point that is made by the authors regards the number of cell types in the avian retina. Based on previous reports in other species, this number may be expected. But how much of the diversity one can observe in amacrine and bipolar cells is related to the transitional states of the same cell type? Also, the authors mention 10 clearly immature clusters: could these correspond to CMZ-derived or CMZ cells? Finally, even if the retina at E18 is histologically mature, some of the cells in mature clusters may be closer to full maturation than others, in particular when it comes to synapse refinement and pruning. Was this taken into account? A first step into this investigation would be to rank the cells with pseudotime alignment method (such as Monocle).

We have responded to the issues of developmental stage and pseudotime analysis above. Unfortunately, we have no data on the CMZ.

2) To what extent is the number of cell types related to i) the number of single-cells sequenced (have you controlled whether you are close to saturation?) ii) the stages used to evaluate them (E12-E18) and iii) the clustering technique (this latter was difficult to assess due to insufficient description)?

We agree that our Materials and methods section was insufficiently detailed, and have expanded it substantially. Regarding developmental stage, the retina is generally viewed as fairly mature at E18, in part because all cells have been born at least 10 days prior to this time. Regarding the number of cells: this is an excellent point, and we had already noted that there may be more types than we identified owing to insufficient sampling. To further illuminate this issue we have now conducted a downsampling analysis of the more diverse classes and provide the results in a new Figure (Figure 12). In brief it suggests have we have captured all or nearly all bipolar and RGC types, but may have missed some rare amacrine types.

3) Why is there no dimensionality reduction visualization of the main dataset? We see these cells aggregated with other species only in Figure 10, but nowhere can we appreciate the distinction between the main classes. Thus, it is hard to assess to what extent these classes are distinct.

We have added a UMAP of the chick data to Figure 1 as requested

4) The manuscript, presented as a resource, occasionally lacks clarity in its presentation, especially in the first pages of the Results section (Figures 1 and 2).

We have revised the text for clarity but note that reviewer 2 found it to be “well written.”

5) The main concern is the lack of detail in the Materials and methods section, in particular on the single-cell analysis (QCs, etc). The clustering appears to be correct, but it would be valuable to add how filtering and hierarchical clustering were performed. Also, a clarification of the rationale for the maximisation procedure used to distinguish cell types would be useful. Same for the thresholds used.

We have expanded the Materials and methods section as requested.

6) Regarding the data availability, the link provided does not work:"https://singlecell.broadinstitute.org/single_cell/study/SCP1159"

The link is set to be private for now but will be activated as soon as the paper is accepted for publication. This is fairly standard procedure.

7) The Introduction and Discussion are well written but poorly referenced, especially when it comes to the multi-species comparison. As this study adds to an increasing body of work on single-cell retina, it seems important to refer to the previous work (unless the number of references is limited?).

We have added additional references, and also cite a comprehensive review now in press in Annual Review of Vision Science (Shekhar and Sanes, “Generating and using transcriptomically based retinal cell atlases”).

Reviewer #3:[…] 1) Throughout the paper the authors associate the concept of single-cell cluster with the concept of cell type. Given that the number and size of clusters can be dynamically changed depending on analysis parameters, I would strongly recommend that the authors avoid using the term "cell type", except where this is well supported experimentally (e.g. by demonstration of a cell-mosaic arrangement of marker expression). This term is especially misleading in the title, which might more accurately be "a single cell atlas of the developing chick retina".

We have changed the paper title as requested. It is now “A cell atlas of the chick retina based on single cell transcriptomics.” We respectfully disagree, however, with the suggestion that we abandon the term “cell type.” As noted in our response to major editorial points, the idea of cell types is firmly established in the literature, and we and others have discussed in detail the thorny issue of distinguishing types from states (e.g., Zeng and Sanes, 2017; Yuste et al., 2020). That is not to say it is a fully solved problem – but the term is now in common use in studies of rodents, primates, fish and flies, and avoiding it leads to nothing but confusion. Instead, we have expanded the Discussion to reconsider the issue and make our criteria clear.

2) The description of the scRNA-seq analyses is really insufficient to permit replication or in some cases interpretation of the data. The authors need to give more details with regard to the specific parameters used for each analysis.

We have substantially expanded the Materials and methods section to provide these details.

3) The authors should provide additional evidence that comparing cell classes and types between the embryonic chick retina to that of the more mature mammalian retinas is experimentally justified. Could it be that they are seeing a greater diversity of cell types in the chick retina because they are considering a greater diversity of transitional states? These multi-species comparisons would be greatly strengthened by addition of a dataset from the mature chicken retina.

We believe that adding a dataset from adult retina is beyond the scope of this study. At a practical level, we do not have institutional permission to obtain or maintain post-hatching chicks on our campus.